# Landscape Features of Costal Waterfronts: Historical Aspects and Planning Issues

**Donatella Cialdea** 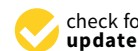

L.a.co.s.t.a. Laboratory, University of Molise, Via De Sanctis, 86100 Campobasso, Italy; cialdea@unimol.it;
Tel.: +39-0874-404-970

**Abstract:** This paper investigates the relationship between different factors that impose on the productive and settlement structures on coastal areas through an analysis carried out on the Italian Adriatic Sea coast. In the panorama of medium- and small-size cities, the relationship between the city, the territory, and the sea very often plays an important role. The main issue of this article is to expose a methodology developed for the definition of landscape quality objectives in the planning of the coast of a region in Southern Italy, Molise. Effort was concentrated on the creation of a territorial survey matrix that could be exploited by local authorities. In drawing up the criteria on which to base the New Regional Landscape Plan, this study provided for the recognition of the identifying matrices for landscape interpretation, creating a database organized in five resource systems. For each resource system, three basic grids were created: each of them collects and processes different information series. These three grids were useful for defining the new protection that is proposed for the sample area. Different conditions emerge in this area, in which two coastal strips have been identified, to the east and to the west of the historical centre.

**Keywords:** landscape regeneration; urban built environment; identity values; smart and resilient land; Natura 2000 Network

## 1. Introduction

The waterfront design and the coastal areas planning are topical issues. The international debate has shown that the potential urban quality of the waterfront in many cases is not in itself sufficient to guarantee an optimization of resources and the management of territorial transformations.

The remarkable interest in urban and territorial planning and regeneration has developed over time: many national and international researchers have defined numerous principles and guidelines on the theme of the coast and of resilient waterfronts in order to adapt them to ecological and sustainable growth and to new transformation models. Some general principles are evoked in order to underline that port cities everywhere are protagonists of major changes, both in terms of physical change and in terms of change in social tissues. Moreover, the specificity of European cities offers a panorama of similar solutions: the coexistence of strong historical references makes theories and application very comparable to each other, especially for the cities that are located in the Mediterranean basin.

Many regeneration interventions involving port areas have proved to have an important role in defining a smart sustainable development model. Starting from local cultural resources, they often manage to combine economic activities of the port with a regeneration of cultural heritage, also through the creativity of their inhabitants [1,2]. Many European cities have done so. Just think of some Spanish cities, like Barcelona or Valencia; some English cities, like Liverpool or Glasgow; some cities in the Netherlands, like Rotterdam or Amsterdam; some French cities, with the important case of Marseille; or even some German cities, with the well-known case of Hamburg. These examples concern cities in Europe. They highlight a sort of "European specificity" for which the new regained waterfronts tend

to combine architectural boldness with territorial interventions, able to create a new face for the cities as a whole [3]. The policies of urban regeneration, in the final analysis, have taken on different faces over the last thirty years, but they shared the enhancement of the common heritage of the city.

In particular, the cases mentioned are apparently similar but instead represent very different urban planning solutions. The continuous interest of urban planning which, in recent decades, has thoroughly examined the issues pertaining to the "waterfront renaissance" [4–7] concerns every waterfront regeneration processes. In the regeneration interventions, the recovery of the built compared to land consumption not only is a physical and environmental recovery but also acts on the local economy; on the quality of living; on the social integration of its inhabitants; and, not less important, on financial instruments related to feasibility, costs, and benefits. The well-known transformations of Barcelona, for example, started as early as the end of the 1980s, were strongly characterized by the impulses linked to the great events [8].

Another example may be the city of Valencia, where the uses of the coastal strip historically had already undergone strong variations over time due to the construction of the port and later for its extension. The port area, in fact, has invaded valuable agricultural areas, with ancient orchards, and above all, it had never built a dialogue with the urban part of the city [9,10].

The case of the port city of Bilbao, where regeneration has involved the course of the Nerviòn river, especially in the part of its estuary, is also a very well-known example. Its transformation from a place of iron and steel business to a new urban reality is interesting from many points of view. It is particularly attractive due to the territorial dimension of urban regeneration policies, which are well integrated with the various intervention scales [11].

The Bilbao metropolitan strategic plan [12], definitively approved in 2003, includes the 34 municipalities of the metropolitan area, which develops along the course of the Nerviòn and includes a population of about one million inhabitants. The purpose of the strategic plan is to carry out a series of interventions, divided into "Aciones Estructurantes" and "Operaciones Estrategicas", which will transform the Bilbao area into a district of advanced services in a modern industrial region.

The case of Marseilles is also an opportunity for reflection: the work of waterfront redevelopment starts from the Euroméditerranée projects and the European Capital of Culture 2013, which aimed to reconnect the city to the sea and to heal a huge rift generated between these two elements when the port was moved to the northern part of the city. Historically, the city is divided into two parts because of the deep bay where the port was present. Marseille is a city that has never completely separated from the sea: it has rather incorporated it and tamed it, making it almost unrecognizable from the land in the cove of the Vieux Port.

Consequently, the need was felt to give continuity to the two shores of the Vieux Port area. A big part of the interventions, in fact, focused on the Vieux Port, classified as a UNESCO heritage site, and on the abandoned area of the J4 pier. Norman Foster won the international competition for the redevelopment and semi-customization masterplan of the Vieux Port in Marseille announced by Marseille Provence Métropole in 2010 [13,14].

Many local administrations prefer to rely on a model already experimented elsewhere and considered a winner: that of the great polarizing building of the ludic-cultural activity of the entire area concerned. This strategy, called the "Bilbao effect", involves assigning great names in architecture, able with their prestige to automatically confer a good degree of legitimacy on the regeneration operation. However, this operation is not always enough in itself to reestablish the authentic bond between the sea and the city. A reconstruction of the relationship between city and port must be concerted with all the actors involved such as local administrations; port authorities; and, above all, the local populations who live this relationship every day with its strengths and its criticalities.

This work analyses the context of major changes in the coastal territory and creates a system of territorial investigation that takes into account all transformations. Environmental, natural, and historical-cultural elements are threatened by growing anthropization, which accurate landscape planning could help to safeguard. Regarding the multiplicity of factors involved, the final goal is to

give a contribution to the definition of how the productive economic system is able to construct new geographies in the panorama of national landscape features. The coastal systems are the ones that are most present in the Italian peninsula and that enrich the number of its geographies, also related to changes in land use over time.

The research background undoubtedly includes multiple activities, which naturally vary from context to context, especially in relation to the economic value of the coastal territories. What this research seeks to highlight, in order to offer new insights, is the desire to involve a framework, the one used in the landscape plan setting, to create a methodology based on territorial analysis, which also supports the choices for the waterfront. The identification of resource systems as an opportunity to compare the values of the area under study is considered the innovative element of the work. The final goal was the collection and subsequently the creation of a Geographical Information System of territorial governance tools that affect the analysed process with the help of GIS software. In this way, data processing and territorial analysis aimed at identifying critical issues and possible urban regeneration scenarios.

This is the result of previous and ongoing research, conducted in the L.a.co.s.t.a. Laboratory (**L**aboratorio per le **A**ttività **CO**llegate allo **S**viluppo **T**erritoriale ed **A**mbientale) of the Molise University, which is at present responsible for the drawing up of the new landscape plan of the Molise Region due to the Agreement between University and Region, who funded this work.

Recognition of landscape protection interventions, oriented to preserve landscape values in harmony with planning tools, have been carried out. Moreover, different productive features, mainly in areas transformed from unproductive to agricultural use, through extensive reclamation operations were focused. The regional territories have been the object of settlement projects related to both tourism and production development. In continuation of what has already been elaborated over the last few years—by virtue of the financing of international projects using funds from Community Initiative Programs [15–17]—the work explores these issues in particular from the landscape point of view, following dictates of the Italian recent law, named the "Urbani Code" [18].

This study is structured in four sections. The introduction (Section 1) has described the main issues of the paper. Section 2 describes the methodology, which also includes a review of methodological approaches for landscape analyses. The next sections (Section 3 with the results, Section 4 with the discussion, and Section 5 with the conclusions) contain the description of the experimental results and comments, geared towards the extension of this work for future research.

## 2. Materials and Methods

The methodology implemented for the New Regional Landscape Plan preparation was oriented to searching the relationship between territory and production processes.

The figure below describes the methodology applied in the different phases of the work (Figure 1).

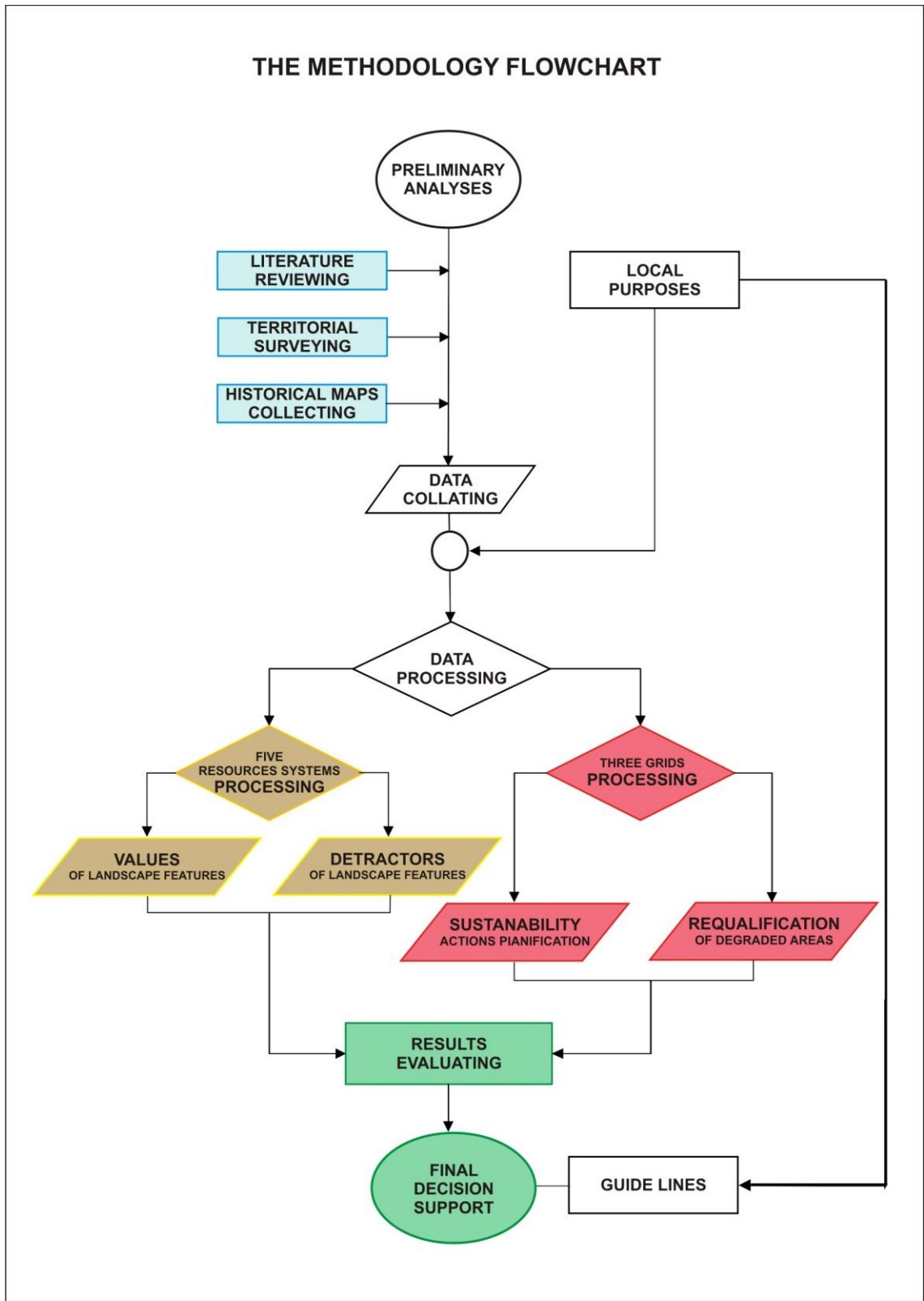

**Figure 1.** The flowchart illustrating the methodology (Source: l.a.co.s.t.a. Laboratory 2018).

*2.1. Preliminary Analyses*

The preliminary analyses, as highlighted in the figure, include the stages of literature review, territorial surveying, and the creation of thematic maps. The first phase includes the process of

drawing up documents deriving from the analysis of the waterfront regeneration examples and from the European documents. In fact, over the years, European legislation has paid increasing attention to the city and to its need for renewal.

In 1975, the "European Charter of the Architectural Heritage" was drawn up, aimed at protecting the heritage through the cooperation of the States (members and nonmembers) of the European area [19]. The document also reflects the emerging interest in environmental issues, as it identifies protection as a tool that has the potential to combat soil waste and the occupation of new agricultural land. In the following years, attention was focused on the "rebirth of the city" and the themes supporting the initiatives to improve the environment were defined, also from the social point of view for the new quality of life. In the mid-eighties, guidelines were beginning to be defined for sustainable development, "a development that meets the needs of the present, without compromising the ability of future generations to meet their own needs" [20].

In 1990, the document known as "Green Paper on the Urban Environment" took shape. The document "promotes the arrest of urban expansion by rejecting the principles of functionalist urbanism, encouraging instead the multi-functionality and a development of densification of the existing" [21]. It also focuses on the problem of urban pollution and identifies actions to stem the phenomenon.

You must get to the mid-nineties with the Aalborg Charter [22] to see definitively "abandoned the functionalist approach proclaimed with the Charter of Athens (1933) from the Modern Movement. The new urban ideology replaces the concept of zoning with that of multi-functionality; the extension of the city with densification; the operations tabula rasa with rehabilitation operations or renewal; the technical and sectorial approach with the partnership approach" [23].

The policies for the city, born in the mid-seventies, received an ulterior impulse from the report on sustainable cities in 1996 [24]. This document promotes the sustainability theory by establishing a series of principles (urban management, political integration, principle of ecosystems, cooperation, and partnership) to pursue and evaluate sustainability strategies and policies in urban areas.

In the following years the debate deepened, up to the drafting of the framework of action for sustainable urban development in the European Union [25]. With this document, the European Commission defines the modalities for the allocation of structural funds for the "urban areas in difficulty" where there is a strong urban decay.

The Hanover conference of 2000 clearly introduces urban regeneration [26] of which the declared objectives are "social justice, reduction of poverty, and social exclusion and a more liveable urban environment" to be achieved through the construction of an economy that is more equitable socially and that is capable of protecting the environment.

Subsequently, in 2004, the document Aalborg + 10 was prepared, according to the guidelines dictated by the Conference, developing the well-known 10 topics [27]: the primary objectives were oriented to revitalize and requalify abandoned or disadvantaged areas, applying principles for sustainable design and construction.

In May 2007, the Leipzig Charter came to life. This document sanctions the need to resort to political strategies aimed at integrated urban development (as a global approach for suitable urban development). It defines concrete development objectives for the urban area and develops a "vision for the city" through action strategies for the "integrated urban development policy" [28].

*2.2. Database Creation*

The study, which provided for the recognition of the identifying matrices for landscape interpretation, led the definition of indicators related to different resource systems. Moreover, the criteria for the selection of the indicators—which will be useful for an evaluation of transformations through time of the territories being studied—has been defined. Five resource systems have been selected. They are physical-environmental; landscape-visual; historical-cultural; agricultural-productive; and demographic-tourism (Table 1).

**Table 1.** The five resource systems.

| Resource Systems | Indicators | Sources |
| --- | --- | --- |
| Physical-Environmental System | The first System includes the indicators relating to Climate and Atmospheric conditions, Water and waterways, and Marine and coastal environments. | The indicators selected derive from different sources: the Council Department for Public Works of the Molise Region, the Arpa Molise, and the Consortium for the Industrial Development of the Biferno Valley. The analysis of the data, which can be found in historical series, were compared with information obtained from soil science and geological maps as well as from geomorphologic maps. Maps of hydrological restrictions were consulted and a map of environmental risk related to landslide and hydrological risk based upon the most recent regional studies was produced. |
| Landscape-Visual System | The second System aims at defining the distinctiveness of the territory. | The morphology of the land was examined through an interpretation of the values already attributed to them in current landscape plans. A map was produced of the officially recognized natural ecosystems based upon the SCI sites (Sites of Community Importance) identified by the Natura 2000 Network. Further information was obtained from Regional Vegetation Maps and from maps drawn up by the Regional Administration of Corine Land Cover level IV soil use. |
| Historical-Cultural System | The third System was analysed through the investigation of elements and areas subject to historical restrictions, through an identification of building typology and through an analysis of landscape visibility. | The analysis of areas subject to restrictions was made by studying each protected historical building and archaeological site (National Cultural Heritage Ministry: historical building and archaeological site and update). The systems of buildings were analysed with an emphasis on the different typologies such as historical centres, rural buildings, towers and coastal defence systems, buildings that were a product of land reforms, large estates, post-earthquake reconstructions, monastic and religious buildings, buildings linked to cattle-tracks, and buildings linked to waterways. |
| Agricultural-Productive System | The fourth System, related to productive-agricultural resources, aims at defining the functions of agriculture. This entails an analysis of the land areas and the fruition of the land in agricultural terms (based upon local council indicators as well as indicators based upon the presence of farms). | All activities linked to agriculture were examined: the traditional farm type, the industrial type, and agricultural tourism. Particular attention was paid to irrigation, given that the coastal areas, as well as the pre-coastal strips, are major areas of irrigation. Information derives from the Historical National Agriculture Census and update, especially for irrigated areas, in the coastal and pre-coastal strips. |
| Demographic-Touristic System | The fifth System was analysed, following the subdivision of local township indicators. These indicate demographic changes, including changes in the farming population, which were then compared to specific indicators linked to industrial activity. | Data derive from the Historical National Population Census information and subdivision of local townships indicators; Local Council Urban Planning Tools and update. Verification of local council urban planning tools was included, paying attention to the large infrastructures foreseen, as these are responsible for major landscape variations, particularly those linked to the sea, ports and inter-ports as well as land communication systems, whether these include further development of pre-existing systems or the creation of new infrastructures. |

### 2.3. Data Processing

The prevailing perspective of the new plan was to improve the protection of the territory's assets—refining the setting of the landscape value—and, above all, to enhance the potential of productive areas in the context of sustainable development of the Molise region, to which, moreover, give an innovative impulse. For this reason, the "values" have always been examined in relation to the presence on the territory of "detractors" who in fact change the value itself. These "detractors" are elements—infrastructural, industrial, or energy related—that invaded the territory over time, creating an uncontrolled development of the territory itself. The territorial analysis was carried out according to the criteria established in the protocol defined in agreement with the Molise Region [29–31]. The collected data was entered into a database, as shown in Table 2.

The sources of ancillary data are shown in the table. Obviously, these data came from heterogeneous databases concerning the various sectors of land use—from the urban sector to the agricultural sector and again to the industrial sector. They have been validated, updated, and subsequently processed to obtain useful information to be included in the Geographical Information System, exclusively created for the Molise Region. The cartographic basis for the territorial analyses is the Regional Technical Map in scale 1:5000, created in the nineties by the Cartographic Research Center of the Molise Region. It was updated for this work, having proceeded to purchase a series of photographic and satellite images (Landsat TM, Spot, and Quick Bird) from the mid 90s to present day; particular attention was paid to data by the MIVIS (Multispectral Infrared and Visible Imaging Spectrometer) sensor, appropriately corrected radio-metrically and, above all, geo-coded (orthorectified)) and georeferenced.

An innovative approach was set up too. For each resource system, three basic *grids* were created: each of them collects and processes different information series. In fact, in addition to the spatial dimension, a time horizon has also been introduced in territorial analyses. The data were grouped according to interpretation that take into account the following:

- for the first grid, the landscape assessments according to the criteria of the old landscape plans (it was called "grid" *A* actual state);
- for the second grid, the evolution of land use over time, from the mid 1950s (it was called *grid E* state evolved over time);
- the third grid analyzes the inconsistencies that derive from the dictates of the urban planning tools of municipalities in order to take into account homogeneous territorial units on a larger scale (it was called *grid P* previsional state).

The availability of all information, already accessible in a GIS environment, allowed further integration of the information in a much simpler and more effective way than would have been possible. It was useful to consider all the information available as a global mass of integrated ancillary data to the point of generating new knowledge. However, the use of the information contained in all these maps has been implemented through data verification and updating.

New information obtained by consideration and rations that emerged from the system of correlations and information queries in the new Geographical Information System were therefore classified.

In drawing up the criteria on which to base the New Regional Landscape Plan, the territorial elaborations carried out for the previous landscape plans of the region were also taken into account. They too were built in the late nineties and constitute an important starting point, even if not complete since the eight floors still in force do not cover the entire regional territory. Since the new legislative provision provides that, instead, the landscape plan involves the entire regional territory, they were found for the missing areas and, for the areas where the data had already been collected and the maps created, they were verified and update. A grids' comparison was made, and some images are shown in the following paragraph that will lead to the definitive conclusions.

**Table 2.** The data collection for the landscape analyses.

| Resource Systems | Values | Detractors |
|---|---|---|
| **PHYSICAL-ENVIRONMENTAL SYSTEM *1**<br>**\*1 Sources**: Molise Region, Department for Public Works; Arpa Molise; Consortium for the Industrial Development of the Biferno Valley, historical series and update. | **1 WATERBODIES**<br>-River Basin<br>-Artificial Lake<br>-Natural Lake<br>**2 NATURA 2000 NETWORK**<br>-SCI (Sites of Community Importance) and SPA<br>-(Special Protection Areas)<br>-Management Plan<br>**3 WOODLANDS**<br>-Forests<br>-State Forests<br>-Nature Reserve<br>-Oasis<br>**4 GEO-SITES** | **1 SEISMIC CONSTRAINT**<br>(zone 1, 2, 3, 4)<br>**2 HYDROGEOLOGICAL CONSTRAINT**<br>**3 HYDRAULIC SETUP**<br>**4 SLOPE STRUCTURE**<br>**5 L 445/1908**<br>Inhabited areas to be consolidated and/or transferred<br>**6 LACK OF MICROZONATION STUDIES** |
| **LANDSCAPE-VISUAL SYSTEM *2**<br>**\*2 Sources**: Molise Region, current landscape plans; SCI sites (Sites of Community Importance) identified by the Natura 2000 Network; Regional Vegetation Maps updated to Corine Land Cover level IV soil use and update. | **1 RESTRICTED AREAS**<br>-Under Law N°. 1497/39<br>-Ope_Legis<br>-State property Areas<br>**2 SIGNIFICANT ELEMENTS**<br>**3 NATURALISTIC VALUES**<br>-SCIs SPAs IBAs Areas<br>-Established Protected Areas<br>-Proposed Protected Areas<br>**4 RESIDUAL AREAS**<br>-Dunes<br>-Civic Uses<br>-Forests<br>-Cattle-tracks Pathways<br>-Pastures | **1 WIND-POWER PLANTS**<br>-Carried out<br>-Planned<br>**2 GROUND-MOUNTED PHOTOVOLTAIC PLANTS**<br>-Carried out<br>-Planned<br>**3 INFRASTRUCTURE NETWORK**<br>-Roads<br>-Railways<br>-Power Lines<br>**4 QUARRIES**<br>**5 LANDFILLS**<br>**6 TREATMENT PLANTS**<br>**7 DEWATERING PUMPS**<br>**8 INDUSTRIAL CENTRES**<br>**9 OTHER INFRASTRUCTURES**<br>-Airports<br>-Ports<br>-Interportsdata |
| **HISTORICAL-CULTURAL SYSTEM *3**<br>**\*3 Sources**: National Cultural Heritage Ministry: historical building and archaeological site and update. | **1 ARCHAEOLOGICAL VALUE**<br>-Restricted Areas under L 1089/39<br>-Areas certified by well-known researches<br>-Areas certified by field surveys<br>**2 ARCHITECTURAL VALUE**<br>-Restricted Areas under L 1089/39<br>-Areas certified by well-known researches<br>-Areas certified by field surveys<br>**3 URBANISTIC VALUE**<br>-Restricted Areas under L 1089/39<br>-Areas certified by well-known researches<br>-Areas certified by field surveys<br>**4 PATRIMONIAL VALUE**<br>-Restricted Areas under L 1089/39<br>-Areas certified by well-known researches<br>-Areas certified by field surveys | **1 ARCHAEOLOGICAL VALUE**<br>-Land ownership<br>-Open Sites<br>**2 ARCHITECTURAL VALUE**<br>-Conservation Status<br>-Restoration Projects<br>**3 URBANISTIC VALUE**<br>-Recovery Plans<br>-Regeneration Plans<br>**4 ACCESSIBILITY TO SITES**<br>-Presence of Highways<br>-Presence of State Roads<br>-Presence of Province Roads |

**Table 2.** *Cont.*

| Resource Systems | Values | Detractors |
|---|---|---|
| **AGRICULTURAL-PRODUCTIVE SYSTEM*4**<br><br>**\*4 Sources:** Historical National Agriculture Census information and subdivision of local townships and update, especially for irrigated areas, in the coastal and pre-coastal strips. | **1 TERRITORIAL SURFACE (TS)**<br>-Municipality TS (ha.)<br>**2 FARMLAND SURFACE (FS)**<br>-hectares<br>-% ST/FS<br>**3 UTILIZED AGRICULTURAL SURFACE (UAS)**<br>-hectares<br>-% UAS/FS<br>**4 BIOLOGICAL CROPS SURFACE**<br>-If > 10% UAS<br>**5 FARMS**<br>-Small < 10 ha.<br>-Medium sized 10–50 ha.<br>-Large > 50 ha.<br>**6 AGRI-FOOD SPECIFICITIES**<br>-D.O.C. (Quality Legislation for Controlled Origin)<br>-D.O.P. (Protected Designation of Origin)<br>-I.G.P. (Protected geographical indication)<br>**7 RECLAMATION CONSORTIUM** | **1 INDUSTRIAL DEVELOPMENT CONSORTIUM**<br>**2 INDUSTRIAL DISTRICT**<br>**3 PRODUCTION PLANS (PIP)**<br>**4 TOWARDS ALTERNATIVE ENERGY**<br>Biogas Production<br>-Forest Biomass<br>-Wood-Chip<br>-Wood chip heating systems<br>**5 OIL MILLS**<br>**6 SUNFLOWER DISTRICT** |
| **DEMOGRAPHIC-TOURISTIC SYSTEM*5**<br><br>**\*5 Sources:** Historical National Population Census information and and subdivision of local townships indicators; Local Council Urban Planning Tools and update. | **1 PLANNING TOOLS**<br>**2 DEMOGRAPHY**<br>-Resident population (2011)<br>-Demographic Trend 2010 (+ve/-ve)<br>-Age classes<br>-% 0–18 years<br>-% 19–40 years<br>-% 41–65 years<br>-% over 65 years<br>**3 ACCOMMODATIONS FACILITIES**<br>-Hotels/Camping/Tourist Villages/Rental accommodation<br>-Farmhouse accommodation/Country-Houses/Youth hostels/<br>-Holiday homes/Bed & Breakfast | **1 PRESENCE OF ASBESTOS**<br>-No sites<br>-Class 1<br>-Class 2<br>-Class 3<br>-Class 4<br>-Class 5<br>**2 WASTES (QUANTITY IN TONNES)**<br>-Waste Recycling<br>-Co-mingled Waste Collection<br>-Total<br>-Non-hazardous waste recovery companies<br>-Landfill site<br>**3 SEWAGE TREATMENT PLANTS** |

## 3. Results

On the specific case of the city of Termoli and, in particular, on its coastal territory, the overlapping of the three grids has made it possible to highlight the need to intervene to safeguard the landscape with differentiated suggestions in the different parts of the municipal territory. Its condition is comparable to other port cities along the Adriatic Sea and to numerous European waterfront situations.

Termoli is the biggest coastal town in Molise, on the Adriatic Sea coast. This case-study provides a starting point for the theoretical discussion. It was examined because it is of particular interest due to two factors: its dimensions and characteristics. The first condition is that it is a significant medium-sized city; the second is that its condition is flanked by the history of a troubled and undeveloped port. In the discussion, it was necessary to have a broader territorial outlook: the Molise coast, in fact, lends itself well to an analysis of production systems for many reasons because many similar phenomena have occurred over time. The most important variations concern the physical conditions due to erosion on a

coast that is generally low-lying, with the exception of short stretches of the high coast of the town of Termoli, which stands on a promontory.

However, along the coast, there are some very important elements, such as the port facilities, including those of a tourist nature, the industrial core, and the large infrastructures, which are also the only significant ones in the whole region, and even several tourist settlements [32–34]. The city of Termoli has the oldest port of the region; recently, a further intervention was carried out on the near recent tourist port of Campomarino and a tourist port was also built in the municipality of Montenero di Bisaccia (although subject to controversy and contrary opinions). On the area immediately behind the coast, there is the Termoli Industrial Centre, undoubtedly the most significant in the region. Located close to the city in the Rivolta del Re district, it is home to the major industries in the area. Its presence since the early 1970s strongly determined a new layout of the territory with the increase in the demographic movement that involved also their surrounding area. Another significant phenomenon was that of land reclamation operations that had various evolutions over time, from the first interventions of the 1930s to the most significant ones of the 1950s by the Apulia, Lucania, and Molise Regions Reclamation Authority. Furthermore, there are a lot of tourist settlements, in building types such as small villas or small owned houses, with widespread phenomena in the 70s and 80s. Finally, behind the small strip of beach, there are the three main infrastructures of Molise, namely the A14 motorway, the State Road No. 16, and the railway. Going through three routes almost always parallel to one another, these infrastructures mark the entire coast.

The Molise Region is characterized by a low population density (about 70 inhabitants/km$^2$), a condition which, together with the difficult geomorphological features, prevented a strong urbanization. Only the coast (about 35 km) has population densities comparable to those of the neighbouring regions. Over the last few years, the city, albeit with a noticeable slowness due to the economic contingency, has undergone some transformations: the construction of a tourist port; the new planning of the port area; the integrated urban development projects involving the seafront and connections between the city and the port; and, finally, the provision of a commercial interport, a new connection between the city and the industrial district, located in the pre-coastal area.

The grid technique takes place by working on territorial analysis, conceived in vector format (which is well suited to the processing of discrete data, collected from the databases described above, very often available on a territorial basis), which was transformed in georeferred raster in order to ratify data deriving from unhomogeneous datasets [35–37].

The process led to defining the final synthetic value of orientation with respect to the objectives of landscape quality through a series of intermediate indicators obtained by processing the related data according to the hierarchical structure (system-elements-elementary data).

In this case, in particular, the accuracy level is that typical of 1:25,000 maps which conventionally equals the intrinsic error in graphics (such is considered the margin of error deriving from the pen mark equal to 0.35 mm, which is the equivalent of just under 10 metres in the nominal scale 1:25,000), which it is not possible to go below in traditional maps. However, it must be stressed that the choice of the 10-metre pixel does not degrade the information to a level that is any lower than the least accurate data and, as such, results in being widely usable for territorial analyses, where even 10 metres are below a significant threshold from the moment that they represent only 100 m$^2$.

The first *grid*, called A, that is the reference for the analysis of *the actual state*—as already declared—was developed from the present "Vast Area Landscape Environmental Physical Plan". In this sample is a plan covering the whole Molise's coastal area and describing the zones with elements of value recognised by the landscape plan itself. In fact, it describes the elements of historical-archaeological value, visual value, productive-agricultural value, natural value, and geological risk.

This plan undertook learning and assessment investigations through diversified maps. The map analysed in this sample area is the map S1 ("Map of the Territory's Quality"). It identifies the elements of which the importance means that they are localised and selected, characterised, and given a value.

The elements useful to the realisation of synthetic analysis were divided by category in accordance with the diverse categories of "interest":

✓　elements of historic, urban, archaeological, and architectural interest;
✓　agricultural-productive elements of interest for natural characteristics;
✓　naturalistic elements of interest for physical-biological characteristics;
✓　areal elements that are geologically unstable; and
✓　elements and environments of visual interest.

　　The grid relative to the "evolved state" (E) was created on the basis of land use: in this sample area, it highlights the main variations with particular attention to wooded areas (in particular, for areas corresponding to reforestation and deforestation), dunal areas (with the aim of defining the zones that have disappeared and those that remain), urban areas (and related progressive expansion), and areas under cultivation (with particular attention for the zones affected by the land reforms).

　　The categories attributed to the polygons for this map have been streamlined and simplified so as to obtain the following keys that are valid for both periods:

✓　urban areas;
✓　agricultural areas (including meadows and meadow-pastureland and arboriculture);
✓　grasslands and wastelands (shrub cover <40% and tree cover <20%);
✓　shrub and bushland (shrub cover >40%);
✓　chestnut plantations;
✓　broadleaf forests;
✓　coniferous forests;
✓　mixed forests of conifers and broadleaf;
✓　reforestation (forestry formation of conifers and broadleaf with h < 5m);
✓　bare ground areas (mountains, coasts, etc.); and
✓　water bodies and wetlands.

　　A comparative layer which analyses precisely the decrease or increase of the land with respect to the sea was introduced. Therefore, the key was structured in the following way:

✓　increase in urban areas;
✓　increase in agricultural areas;
✓　increases in woodland areas;
✓　increase in bare ground;
✓　increase in shoreline;
✓　decrease in agricultural areas;
✓　decrease in woodland areas;
✓　decrease in bare ground;
✓　decrease in shoreline; and
✓　no change.

　　Finally, the grid for the "previsional state" (P) has been derived from current urban planning tools and consists in the identification of the various designated zones with their relative attributes (indexes of suitability for building and of the various designated uses). In this sample area, it consists in identifying the various destination areas with their attributes. The degradation of the natural ecosystems characteristic of the coast has certainly begun in former times, but it has had a great increase in even more recent times, from the postwar period onwards. In fact, the spontaneous, chaotic, deregulated coastal development has pushed the anthropic presence ever closer to the shore. The Italian coast and, in particular, along the Adriatic Sea was affected above all by a strong increase in the construction of second homes right along the shoreline, also contravening the prohibition of

construction for the coastal strip for its 300 m of depth, as was dictated since the aforementioned law on the landscape of 1939 (it is in fact a large part of houses illegally built but is subsequently remedied by building amnesty).

In the landscape analysis of the case study (Figure 2a), the most sensitive zone was deepened. They are, in particular, the MV1 areas (high level in visual values) and the MS areas (moderate visual value). For these areas, a medium sensibility to transformation was enshrined. In the MV1 area (in which the Industrial Area of Termoli is located), almost every use is allowed (under the conditioned transformation mode), except for settlements that are subject to the environmental assessment procedure. For the biggest MS area, with lots of residential settlements, intervention is possible, under the conditioned transformation mode.

This part of the regional coast also includes important areas from a naturalistic point of view: they are those designated as the A2N1 areas, which are strongly characterized by natural elements, and the A2N2 areas, where the natural vegetation is characterized by exceptional visual and naturalistic value (Figure 2b).

Their selection was implemented: in addition to sustainability indicators, other indexes have been added, which are considered important for the assessment of the sustainability of regeneration processes. Moreover, in addition to these indicators derived from the literature and from the regional databases, further specific indicators were identified as necessary to assess the landscape quality, and they have been added to our work. Consequently, elements related to the categories of natural, visual, perceptive, historical, and productive elements have been identified as landscape-related indicators. As regards the relationship between sea and city, studies carried out for the old town area underline some negative aspects, such as the isolation of the old town and the impossibility of enjoying the sea from the town centre. The presence of the sea is only partially perceived through a few viewpoints, and recent building has further contributed to the closure of the view from possible panoramic points.

The two zones of the old town and the 19th century town, which form the heart of Termoli, are in reality the hub around which the entire town's main economic, cultural, and tourist activities revolve. The first obstacle to an organic expansion of the town is formed by the railway, (inaugurated on 1864) which isolates the "19th century town" from the rest of the territory. The second barrier, built around 1960, is the motorway. The Adriatic State road No.16, which runs parallel to the motorway, adds to this series of infrastructural axes. In fact, the zone between the motorway and the Adriatic State road No.16 forms a kind of artificial island.

Therefore, specific analyses were carried out about the current urban plan (Figure 2c). The city of Termoli is the first of the four towns along the Molise coast to have a master plan. The first plan, adopted in 1971, was approved by the regional council on 1972. Three years later, the general variant to this master plan was adopted, and it was approved in 1977. Later, various partial variants have been proposed that have completely modified the plan's original design in order to adapt it to new needs. A general variant was adopted by the town council in 2003, but it was never approved by the region [38].

Really, the plan created a transversal link between the three homogeneous strips parallel to the coast and delimited by the great transport networks:

- the coastal strip (north coast, old town, Rio Vivo);
- the intermediate strip (delimited by the railway and motorway); and
- the external strip (beyond the Road Variant to the State Road No. 16)

However, the determinant factor in the original choices of the plan is the town's vocation to tourism activities and the will to adequately upgrade this aptitude that Termoli has gone.

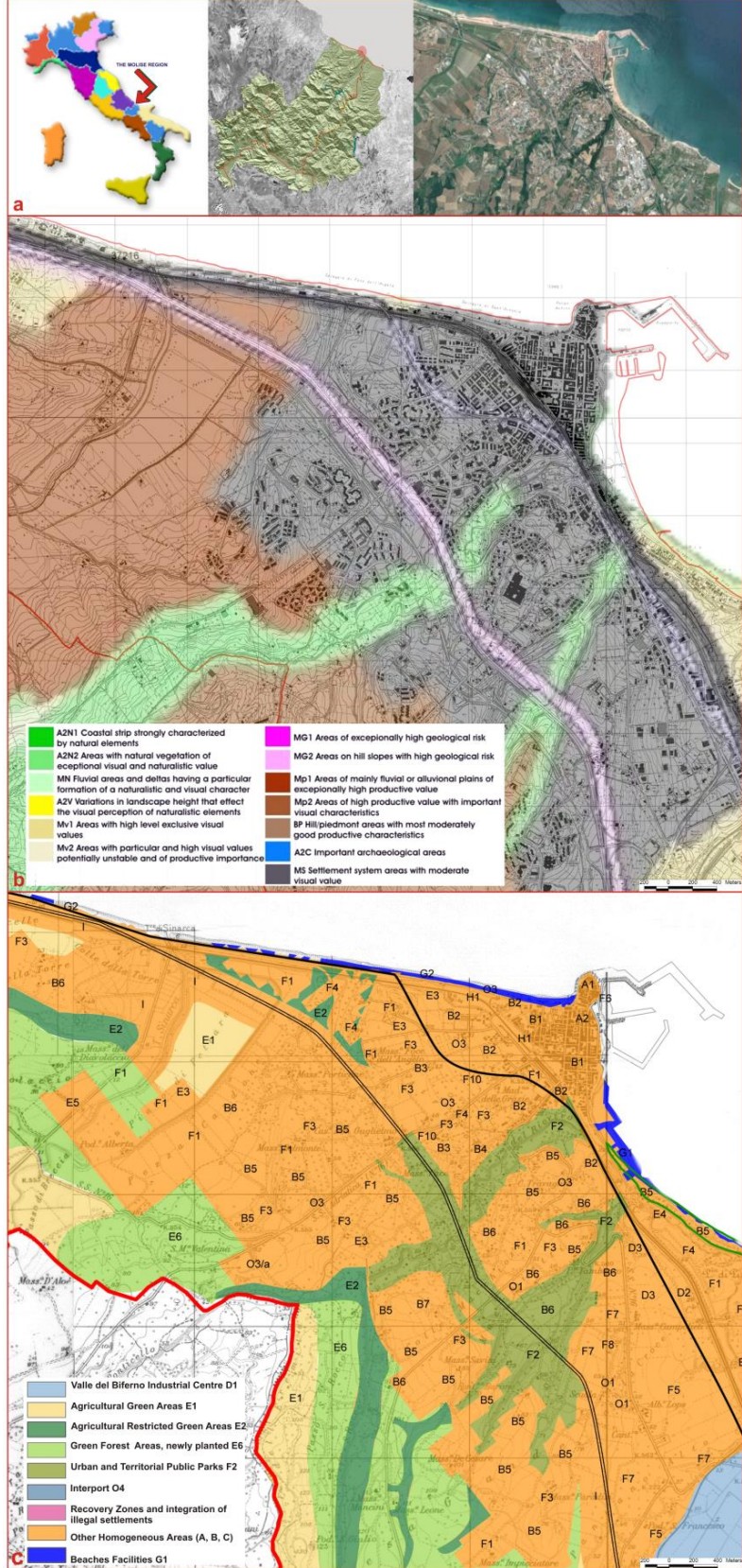

**Figure 2.** The sample area: (**a**) localization and orthophoto with the Molise Region and the Termoli Municipality; (**b**) transformation tools provided by the current landscape plan for the area surrounding Termoli; and (**c**) Termoli master plan and its zoning (Source: l.a.co.s.t.a. Laboratory 2019).

The local plan provides further interventions oriented to a) the valorisation of tourist accommodation and connected structures; b) the organization of artistic, archaeological, historical, and environmental resources; c) the identification of structures for leisure time use; and d) the construction of transport infrastructures and car parks. In the coastal strip, seasonal commercial structures providing services for beach activities are planned. The intermediate strip is planned in order to locate an exhibition-show area with ample spaces dedicated to permanent exhibits of local crafts; furthermore, an office district combining large infrastructures such as spaces for entertainments, offices, and city services is planned. The green area located just south of the park, which is unused and abandoned, in the plan will be turned into a theme park. In the external strip, green areas, services, and perimeters of the zoning are redrawn and new building areas are created.

In addition, some projects have recently been funded and carried out for the urban waterfront. The new waterfront of Termoli comprehends the new master plan of the Commercial and Tourist Port and the urban historical centre regeneration. The new distribution of functions and services improve the relationships between the activities of the harbour and the fisherman village. Moreover, in the Rio Vivo—Marinelle zone, there is the only urban park in Termoli, designed in 1962 and definitively approved in 1964. In the following years, some active sports facilities were added, including tennis and soccer fields, and finally a large swimming pool that is now in disuse. The current Termoli's master plan classified this area as a densely urbanized agricultural area (E1), although in the 1977 variant it was intended for the port equipment (F5). It is delimited to the south by the industrial zone and, more precisely, from the strip to the ever-built interchange to the southeast by the reclamation channel of contrada Marinelle; to the southwest from the road infrastructure (State Road No. 16) and the railway (the so-called Ferrovia Adriatica) and its coastal fragmentation barriers; and to the north by the Urban Beach Plan, which is a sandy beach subject to remarkable erosive phenomena attributable to the port of Termoli. The strategic environmental assessment for the master plan of the port provided for a development scenario until 2025, and it highlighted how the current configuration causes a significant backsliding. Moreover, the area under study is included in site of community importance named "Foce del Biferno—Campomarino Coast", in which there is a strong anthropization, considered the cause of the destruction of the dune, as well as detecting the danger to the pine wood now in contact with the marine waters due to the beach retreat.

## 4. Discussion

Territorial analyses have highlighted, within the study area, zones in which the greatest conflict occurs: for these areas, specific rules for the regeneration policies are necessary [39].

In particular, the municipality examined is located in a region with strong agricultural features: the excessive consumption of agricultural and natural soil not only deeply affects changes in the landscape but also involves problems in ecological assessment, especially inhibiting natural regeneration capacity of environmental resources. Moreover, this is well known to be a more and more of growing risk for urban areas, affected by poor planning choices and filled with many cultural elements [40]. Furthermore, the approach necessarily started from the current settlement assessment, with its economic and environmental conditions, looking for a new ecological situation based on the balance between the environmental resource availability and their uses. In this paper, therefore, the portion of the city of Termoli—which is located within a significant naturalistic and agricultural context—has been examined. In this land portion, it is however necessary to consider the emergent factors of the coastal landscape. Analyses were carried out, taking into account the resilience as a capacity of the system to be able to maintain stability compared to an initial state of equilibrium [41–43].

As well known, the most delicate situations in the analysis of resilience are both in the face of calamitous events, which produce instant change of state, and in the face of slower changes that occur through the growth of the city.

In this case-study, both conditions are true: in fact, significant flood events have occurred in recent years and continuous mutations are producing gradual changes over time [44]. Especially in cases

characterized by prolonged crisis, it is important that systems were able to adapt to preserve their own identity. A major role should be attributed to "environmental infrastructure" and in particular to so-called "blue infrastructures", namely watercourses and coastal waters. Such elements should be attributed to new city development policies.

In order to prepare the new regional landscape plan, a table of the new landscape quality aims as listed in art. 135 of the National Code (L. 42/2004) was organized: it contains for each homogeneous area specific requirements and provisions, oriented to the conservation of the constituent elements; to the rehabilitation of compromised or degraded areas; to the protection of landscape features; and to the identification of the lines for the development, with particular attention to the preservation of rural landscapes and sites included in the UNESCO World Heritage List. Moreover, about the costal area in which our sample test is, the final targets are related to the conservation, protection, management, and planning of exceptional, ordinary, and degraded landscapes with particular reference to typical natural landscapes such as rivers, lakes, hills, mountains, coastal and rural landscapes, forestry, and agro-pastoral, not to mention historic, rural, urban, industrial, and infrastructure sectors (Table 3).

The objectives identified are also related to the government of the processes of urbanization and abandonment of the territory and to the preservation of material cultural values and intangible values such as the traditions and history of the region.

The general objective was subdivided into specific objectives, as shown in the table, in which these objectives were finally associated with landscape quality directions that indicate policies to adopt and those who have an interest in achieving these objectives as well as the measures required to adapt the urban planning instruments to the indications of the new regional landscape plan.

Therefore, the landscape quality targets in this area aim to safeguard the surviving heritage in the area, to recover and improve the landscapes altered and degraded by human activity and to define quality standards.

Subsequently, the three grids, set by this methodology, were useful for defining the new protection that is proposed for the coastal area in the municipality of Termoli (Figure 3). They are derived from the analysis of the tool dedicated to the protection of the landscape (starting from the definition of the current plan), from the city planning tool (master plan of the municipality of Termoli) and from the dictations of the other tools in force in this zone, which is the port master plan of the and the beaches and coasts plan. The diversity of the two areas of the coast clearly emerges, located around the historical urban centre. For these two areas, there is a strong concentration of the "built" but not predominantly "urban": the two coastal strips (zone A to the west and zone B to the east) that surround the promontory where the historic centre is located and includes the port area, which has taken on different and increasingly broad connotations over time (Figure 4).

**Table 3.** The landscape quality aims declined for the five resources systems in the sample area.

| RESOURCESYSTEMS | GENERAL AIMS | SPECIFIC AIMS |
|---|---|---|
| PHYSICAL-ENVIRONMENTAL SYSTEM | **1** **PROMOTE THE PRESERVATION OF THE INTEGRITY OF AREAS OF HIGH NATURALNESS AND HIGH ECOSYSTEM VALUE** | 1.1 Safeguard geological-geomorphological systems with high integrity (geological formations, ravines, cliffs, crags) <br> 1.2 Safeguard protected areas and areas of high environmental value such as those covered by the Nature 2000 Network <br> 1.3 Safeguard and improve environmental functionality of river and lake systems of Molise <br> 1.4 Safeguard and rebuild coastal marine habitats of Molise <br> 1.5 Safeguard woods and forests of mountainous and hilly areas of Molise <br> 1.6 Redevelop and redesign the coastal landscapes of Molise |
| LANDSCAPE-VISUAL SYSTEM | **2** **PROMOTE IMPROVED INTEGRATION OF LANDSCAPE AND THE QUALITY OF INFRASTRUCTURES** | 2.1 Define territorial and landscape quality standards in the settlement of new network infrastructure <br> 2.2 Define territorial and landscape quality standards in the settlement of new energy infrastructure <br> 2.3 Define territorial and landscape quality standards in the settlement of new productive activities |
| HISTORICAL-CULTURAL SYSTEM | **3** **PROMOTE THE PRESERVATION OF CULTURAL VALUES** | 3.1 Preserve cultural value and witnesses of settlements and historical manufacts <br> 3.2 Preserve cultural value of traditional rural buildings <br> 3.3 Preserve the visible cattle-tacks residual <br> 3.4 Redevelop the historic rural landscapes |
| AGRICULTURAL-PRODUCTIVE SYSTEM | **4** **PROMOTE THE CONSERVATION OF AGRICULTURAL LANDSCAPES** | 4.1 Develop the agricultural landscape of Molise, recognize and promote its social functions <br> 4.2 Preserve open landscapes of the reclamation as a characteristic aspect of identity of coastal landscape of Molise <br> 4.3 Redevelop the agricultural landscape of Molise |
| DEMOGRAPHIC-TOURISTIC SYSTEM | **5** **PROMOTE THE IMPROVEMENT OF THE QUALITY OF THE SETTLEMENTS** | 5.1 Improve quality of urban settlements and their environmental performance, for greater well-being of the population <br> 5.2 Redevelop degraded contemporary urbanization landscapes <br> 5.3 Improve urban quality and touristic settlements <br> 5.4 Improve urban quality of agricultural and productive settlements <br> 5.5 Improve soft mobility quality (walking, cycling, trekking on horse) and its interconnection with traditional mobility |

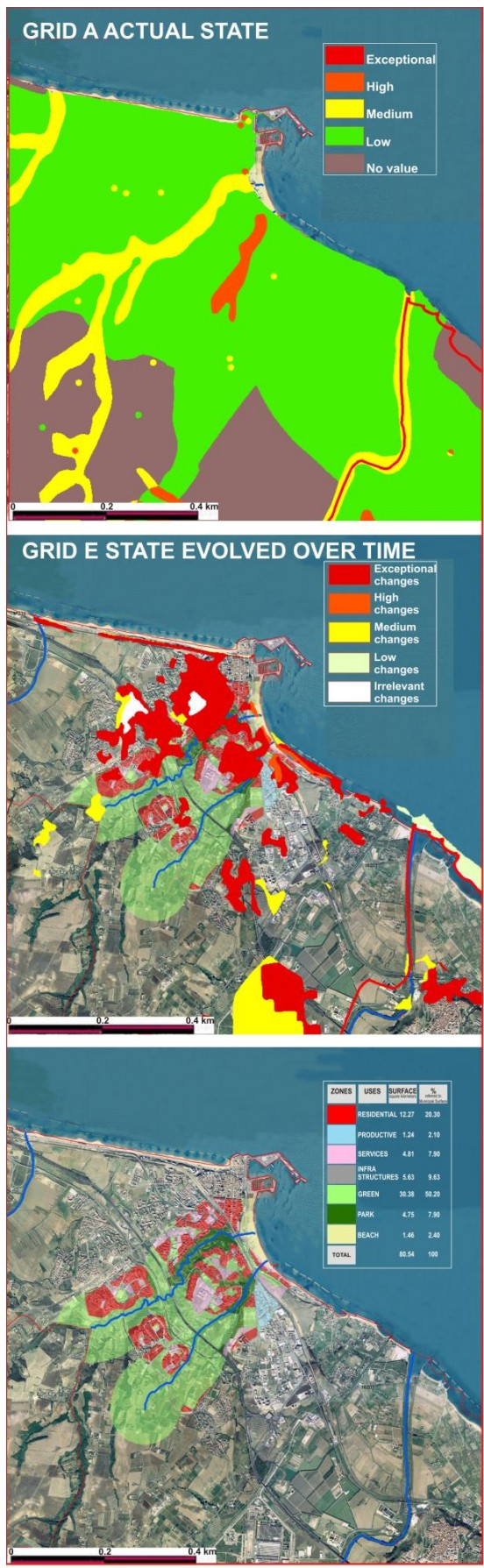

**Figure 3.** The three grids for the Molise coastal zone (Source: l.a.co.s.t.a. Laboratory 2019).

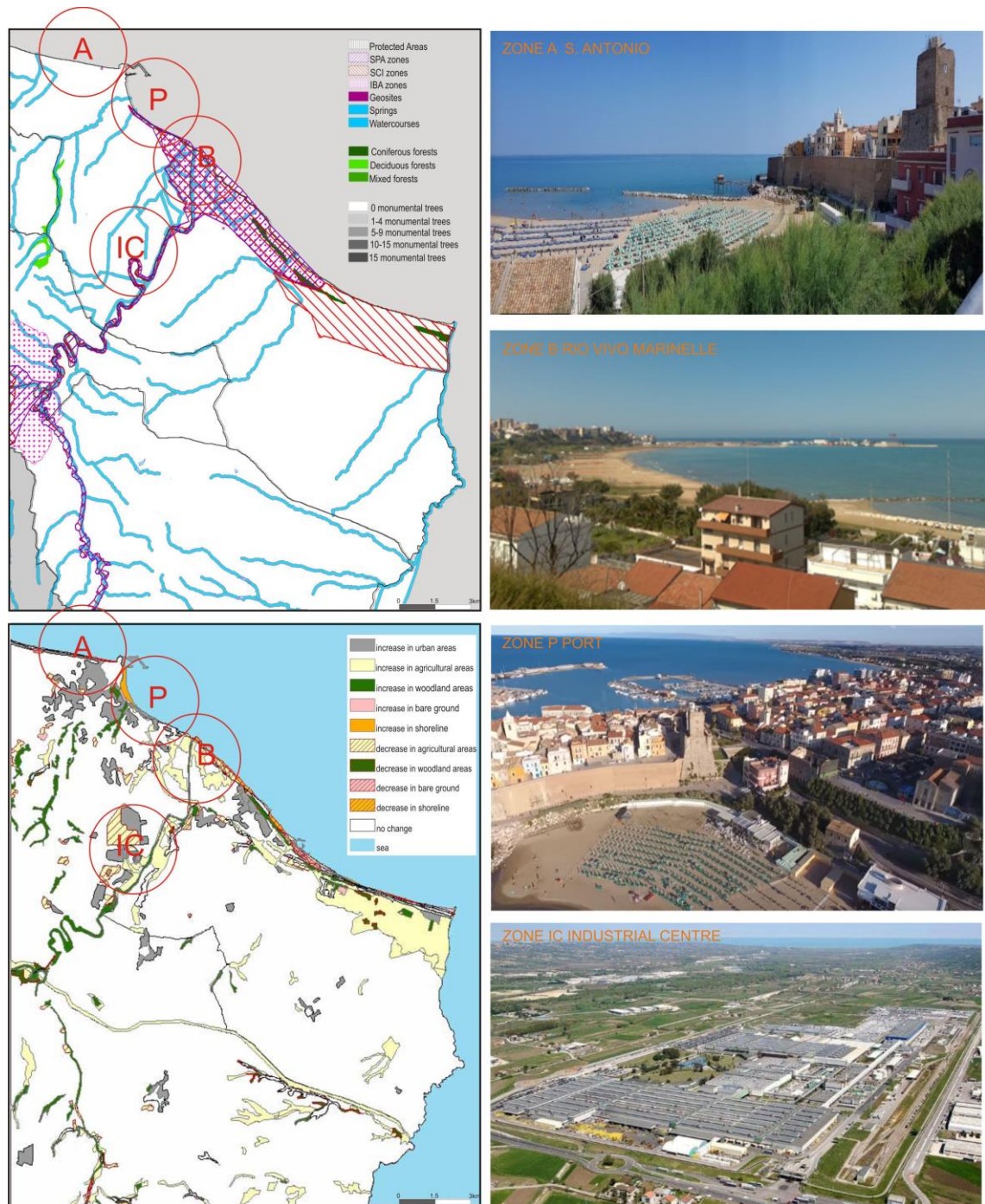

**Figure 4.** The two different coastal areas surrounding the urban centre (Source: l.a.co.s.t.a. Laboratory 2019).

These two different conditions were deepened using the matrices created with the data of the five systems and of the differences in land use in the respective pre-coastal areas.

## 5. Conclusions

The study clearly showed that anthropogenic pressures on the coast are increasingly evident. Even comparing this area to the rest of the regional territory, in fact, the work cannot help but highlight that the greatest transformations, with strong consequences on the landscape image of the territory, took place right on the coast. Obviously, for the constitution of the water basins (located in the pre-coastal

zone), the urbanization occurred quickly and without controls (especially second homes) and new assets related to increasing infrastructures contributed to a strongly incoherent arrangement.

The nonurban spaces are the central focus of attention as they are no longer tied only to agricultural production (that pervaded in the past, especially the pre-coastal area as a result of reclamation operations) but increasingly to the creation of services to the population. In these areas, it is necessary to maintain the quality of the landscape with its identifying characteristics and with its differentiations related to the places' traditions.

On the whole, as far as the analyzes carried out are concerned, the clear agricultural and production vocation emerges: in particular, the lands in the valley and the irrigated ones in the hills are of exceptional value, in light of both the geopedological analyzes and cropland attitudes. No less important is the presence of SCI (Sites of Community Importance) areas that involve, as seen, the entire coastline of the Molise region. The element that has most changed the landscape is the increase in the built environment; furthermore, a substantial change was defined by the construction of the vast industrial area.

The final considerations, therefore, concern above all the destinies of the "built", now considered part of the well-established fabric of the urban areas, even if lacking in specific identifying characteristics.

The under-study areas, however, are clearly demarcated, on one side, by high infrastructure lines (railway, highway, and Adriatic road) and, on the other, by the sea line, as seen invested by sites of community importance areas. The opportunities that the new regional landscape plan can offer, also on the basis of the experiences of the other plans already approved or in the process of being approved in the other Italian regions, can be harbingers of a new philosophy of safeguarding the coastal areas, which does not destroy the existing but is able to restore a "landscape balance".

Coastal planning is a central aspect of this study. In fact, the most critical elements of the demographic and economic development of the regional realities concerned converge on the coast areas. The government of the territory can be implemented through an intervention perspective that concerns a plurality of characters inherent in natural and artificial resources to guarantee protection of environmental and territorial peculiarities. The analysis and comparison between objectives, plan strategies, environmental compatibility criteria, environmental reference indicators, and monitoring methods are all parameters that come into play, but they are not sufficient. Therefore, this vast study is intended to be a contribution to the deepening of the relationship between territorial planning and landscape planning, since the urbanization processes of the coastal areas inevitably influence even the innermost territories, also affected by related economic pressure tourism development that endangers the integrity of the most fragile environmental and landscape contexts. The analysis carried out demonstrates the need to set up a methodology for the landscape which, although starting from data collection based on a homogeneous approach, in reality must always be adapted to local conditions, even within the same region. The final goal is to create vital landscapes.

This study was performed in two different scales: at the territorial level and at the local level, that is, the municipal level. At the territorial level, the indications are intended to address and propose planning actions. At local level, the municipal plans should act in accordance with the landscape plan suggestion: within the areas mentioned above, different situations are identified (in relation to the features and values of the natural environment and human actions) that require specific norms and guidelines. The overlap of the current and forecast condition also allowed us to have an immediate (and quantifiable) perception of dangerous, intense, extensive, and persistent modifications, especially where there is neglect and poor preservation.

The future improvement of the research will be the suggestions for interventions that municipal administrations could promote, as is already happening in some other regions, as active tool for the protection of the landscape within their municipal planning tools, in compliance with what is defined by the regional landscape plan. This tool must naturally understand how the individual management tools for parts of their territories already foresee but further implement them in a more specific logic for urban planning.

**Funding:** This research was funded by the Molise Region, under Agreement with the l.a.co.s.t.a. Laboratory (Director prof. D. Cialdea) of the University of Molise for the realization of the "*New Regional Landscape Plan of Molise*", signed 11.02. 2011, REG. n. 11 del 28.02.2011.

**Conflicts of Interest:** The author declares no conflict of interest.

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
