# Peer review of "Landscape Features of Costal Waterfronts: Historical Aspects and Planning Issues"

_sustainability, doi:10.3390/su12062378_

Round 1
Reviewer 1 Report
It is potentially interesting study, where landscape values are juxtaposed with detractors in a comprehensive way.
However, the work seems not clearly explained, and at times, is challenging to follow authors line of thought (perhaps poor level of English writing is one of the reasons). Manuscript sections are mixed up with each other.
The abstract could more explicitly describe the goal and scope of the study.
Besides the above, more detailed comments are stated below:
- Double-check the citation style: [9,10,11,12] > [9-12].?
- Line 99 - what do authors mean by that from the landscape perspective the "interrelationship between the city and the sea enhances the characteristics of the urban landscape"? What characteristics are enhanced?
- Line 100 - It is stated that the promenade is the opportunity for redevelopment. What is the logic of this statement? Is any space an opportunity of redevelopment?
- Line 114-117 - this sentence seems too long, and unclear.
- Line 133 on - Authors describe several case examples of waterfront regeneration. It could be useful to create a table and/or a figure depicting and comparing the main differences and commonalities of these cases.
- Line 345 - Authors introduce the concept of "New Regional Plan" "new Landscape Plan" and "New Landscape Plan" "Regional Landscape Plan". Do all these refer to the same elaboration?
Line 354 -355 sentence not clear
Line 369 - time periods of satellite images not provided.
Table 1 - The table seem to belong in the result section
- Why Dunes are identified as a visual resource, but not environmental one any more?
- Agricultural fields, and farms in terms of their area are presented here as values. While being a part of the cultural landscape the agricultural sprawl and monocultures are considered a negative phenomenon, a risk to the natural habitats, and changing the ecosystem functioning significantly.
- In the Demographic/ touristic resource system row Planning tools as a value is not clear. Also, it looks like some values are lacking , for example the touristic infrastructure including roads, public transport and so forth. It is unclear how the presence of asbestos directly influences regional tourism.
Majority of the "Results" section seems to belong to the method section.
figure 1 not legible.
Line 569 on - This is discussion
The contributions and limitations of the study have not been clearly described in the Conslusions section.
Author Response
The Author would like to thank the Referees for their comments. The manuscript has been modified following the suggestions, when deemed appropriate (new parts are in green). In this document, please find the details of my responses.
The Author.
REFEREE 1
Comment 1
The abstract could more explicitly describe the goal and scope of the study.
Reply 1
The abstract has been rewritten, emphasizing the goal and scope.
Comment 2
Double-check the citation style: [9,10,11,12] > [9-12].?
Reply 2
Citations have been corrected.
Comment 3
Line 99 - what do authors mean by that from the landscape perspective the "interrelationship between the city and the sea enhances the characteristics of the urban landscape"? What characteristics are enhanced?
Reply 3
The sentence has been rewritten: The city’s ability to interact with the sea increases the landscape value – useful for our territorial analyses - of the urban front.
Comment 4
Line 100 - It is stated that the promenade is the opportunity for redevelopment. What is the logic of this statement? Is any space an opportunity of redevelopment?
Reply 4
The sentence has been rewritten:
Secondly, the regeneration of the waterfront promenade, running along the beach, means that it becomes a place to be lived as a public space.
Comment 5
Line 114-117 - this sentence seems too long, and unclear.
Reply 5
I removed it from the text as, on review, I considered it to be superfluous.
Comment 6
Line 133 on - Authors describe several case examples of waterfront regeneration. It could be useful to create a table and/or a figure depicting and comparing the main differences and commonalities of these cases.
Reply 6
Honestly, I think it could be better to add a sentence explaining that the examples are not meant to be exhaustive for a general overview (for this reason the table would not make sense). The examples were selected due to their suggestions which led the research group to draft the plan for the Molise coast.
Comment 7
Line 345 - Authors introduce the concept of "New Regional Plan" "new Landscape Plan" and "New Landscape Plan" "Regional Landscape Plan". Do all these refer to the same elaboration?
Reply 7
You are right, even if I didn’t find "New Regional Plan". The sentences have been corrected in this way (with lower case when talking generally and with capital letter when referring to our plan):
LINE 42 - NOW LINE 106
new landscape plan of the Molise Region
LINE 299- NOW LINE 69
preparation of the different regional landscape plans
LINE 346- NOW LINE 122
New Regional Landscape Plan
LINE 352- NOW LINE 356
just: new plan
LINE 395- NOW LINE 399
New Regional Landscape Plan
LINE 627- NOW LINE 629
New Regional Landscape Plan
Comment 8
Line 354 -355 sentence not clear
Reply 8
On review deemed superfluous.
Comment 9
Line 369 - time periods of satellite images not provided.
Reply 9
I put it in the text: from mid 90’s to present day.
Comment 10
Table 1 - The table seem to belong in the result section
Reply 10
The table forms an integrated part of the methodology: it is a list of elements included in the database.
Comment 11
Why Dunes are identified as a visual resource, but not environmental one any more?
Reply 11
Dunes are only in the landscape-visual system because they have almost completely disappeared and in our work they are considered only as residual areas and considered as a value.
In the first system (physical-environmental system) we only put the officially recognized values as the environmental values; in the second system (landscape-visual system) a forth category was created as “residual areas” that could improve the landscape value.
Comment 12
Agricultural fields, and farms in terms of their area are presented here as values. While being a part of the cultural landscape the agricultural sprawl and monocultures are considered a negative phenomenon, a risk to the natural habitats, and changing the ecosystem functioning significantly.
Reply 12
If your remark means that extensive agriculture is a negative factor, I agree. However, the extensive areas mentioned in the Molise region are always limited: extensive agriculture does not exist. It is traditionally a highly agricultural region, however based on pulverization and fragmentation characteristics (small farms, even fragmented). It was the topic of my first monograph: CIALDEA D. (1995), Il Molise una realtà in crescita. Aree protette e attività agricole. Ed. Franco Angeli/Urbanistica. Milano. pp. 1-408.
Comment 13
- In the Demographic/ touristic resource system row Planning tools as a value is not clear. Also, it looks like some values are lacking , for example the touristic infrastructure including roads, public transport and so forth. It is unclear how the presence of asbestos directly influences regional tourism.
Reply 13
The system describes elements related to inhabitants, residents and tourists. The asbestos is toxic. The presence of asbestos was found through our processing of satellite images, using the ENVI software.
Comment 14
Majority of the "Results" section seems to belong to the method section.
Reply 14
The description of the three grids is necessary to clearly introduce the results. Nevertheless, in the methodology a flowchart was included.
Comment 15
figure 1 not legible.
Reply 15
I think it refers to the size of the characters in the legend. The image has been redone.
Comment 16
Line 569 on - This is discussion
Reply 16
Ok. Corrections made.
Comment 17
The contributions and limitations of the study have not been clearly described in the Conslusions section.
Reply 17
Ok. Corrections made.

Reviewer 2 Report
The research presented by author has been devoted to an important issue that is particularly topical in coastal areas Adriatic Sea. It is important, comprehensive and new. Relationship between the territory and production processes in this area will be interesting for scientists, PhD students and many other readers. Development of territorial survey matrix for definition of landscape quality objectives in the planning of the coastal areas has high added value for local authorities.The article meets high requirements of the journal, after minor revisions needs to be supplemented by the mentioned remarks and is recommended for publishing.
- It is advisable to explain more detailed why such five resource systems as physical-environmental, landscape-visual, historical-cultural, agricultural-productive, demographic-tourism have been selected.
- What are the consequences of introducing specific indicators, identified as necessary to assess the landscape quality? If there are data, it is advisable to add them.
- It is advisable to explain more detailed an essence of application of matrices created from the data of five systems and differences in land use in respective pre-coastal areas.
- The discussion provides too little information about interaction between non-urban spaces, agricultural production and landscape balance.
- It is important to present the recommendations for future studies in the end of the manuscript.
Author Response
The Author likes to thank the Referees for their comments. The manuscript has been modified following the suggestions, when deemed appropriate (some paragraphs are reorganized: only new parts are in grey). In this document, please find the details of my responses.
The Author.
COMMENT 1
- It is advisable to explain more detailed why such five resource systems as physical-environmental, landscape-visual, historical-cultural, agricultural-productive, demographic-tourism have been selected.
COMMENT 2
- What are the consequences of introducing specific indicators, identified as necessary to assess the landscape quality? If there are data, it is advisable to add them.
COMMENT 3
- It is advisable to explain more detailed an essence of application of matrices created from the data of five systems and differences in land use in respective pre-coastal areas.
COMMENT 4
- The discussion provides too little information about interaction between non-urban spaces, agricultural production and landscape balance.
Reply1 2 3 and 4
A brief explanation of the five Resource Systems has been introduced in the first part of the second paragraph (Materials and Methods).
Moreover, the criteria for the selection of the indicators - which will be useful for an evaluation of transformations through time of the territories being studied – has been defined. Five Resource Systems have been selected. They are: Physical-Environmental; Landscape-Visual; Historical-Cultural; Agricultural-Productive; Demographic-Tourism (table 1).
Table 1. The Five Resource Systems
|
Resource System |
Indicators |
Sources |
|
Physical-Environmental |
The first System includes the indicators relating to Climate and Atmospheric conditions, Water and waterways, and Marine and coastal environments. |
The indicators selected can be obtained from different sources: the Council Department for Public Works of the Molise Region, the Arpa Molise, and the Consortium for the Industrial Development of the Biferno Valley. The analysis of the data, which can be found in historical series, were compared with information obtained from soil science and geological maps as well as from geomorphologic maps. Maps of hydrological restrictions were consulted and a map of environmental risk related to landslide and hydrological risk based upon the most recent regional studies was produced. |
|
Landscape-Visual |
The second System aims at defining the distinctiveness of the territory. |
The morphology of the land was examined through an interpretation of the values already attributed to them in current landscape plans. A map was produced of the officially recognized natural ecosystems based upon the SCI sites (Sites of Community Importance) identified by the Natura 2000 Network. Further information was obtained from Regional Vegetation Maps and from maps drawn up by the Regional Administration of Corine Land Cover level IV soil use. |
|
Historical-Cultural |
The third System was analysed through the investigation of elements and areas subject to historical restrictions, through an identification of building typology and through an analysis of landscape visibility. |
The analysis of areas subject to restrictions was made by studying each protected historical building and archaeological site (National Cultural Heritage Ministry: historical building and archaeological site and update). The systems of buildings were analysed with an emphasis on the different typologies such as historical centres, rural buildings, towers and coastal defence systems, buildings that were a product of land reforms, large estates, post-earthquake reconstructions, monastic and religious buildings, buildings linked to cattle-tracks, and buildings linked to waterways. |
|
Agricultural-Productive |
The fourth System, related to productive-agricultural resources, aims at defining the functions of agriculture. This entails an analysis of the land areas and the fruition of the land in agricultural terms (based upon local council indicators as well as indicators based upon the presence of farms). |
All activities linked to agriculture were examined: the traditional farm type, the industrial type, and agricultural tourism. Particular attention was paid to irrigation, given that the coastal areas, as well as the pre-coastal strips, are major areas of irrigation. Information derives from the Historical National Agriculture Census and update, especially for irrigated areas, in the coastal and pre-coastal strips. |
|
Demographic-Tourism |
The fifth System was analysed, following the subdivision of local township indicators. These indicate demographic changes, including changes in the farming population, which were then compared to specific indicators linked to industrial activity. |
Data derive from the Historical National Population Census information and subdivision of local townships indicators; Local Council Urban Planning Tools and update. Verification of local council urban planning tools was included, paying attention to the large infrastructures foreseen, as these are responsible for major landscape variations, particularly those linked to the sea, ports and inter-ports as well as land communication systems, whether these include further development of pre-existing systems or the creation of new infrastructures. |
Moreover, this explanation was added into the fourth paragraph (Discussion).
In order to prepare the New Regional Landscape Plan, a table of the new landscape quality aims as listed in art. 135 of the National Code (L. 42/2004) was organized: it contains specific requirements and provisions for each homogeneous area, oriented to the conservation of the constituent elements, to the rehabilitation of compromised or degraded areas; to the protection of landscape features; to the identification of the lines for the development, with particular attention to the preservation of rural landscapes and sites included in the UNESCO World Heritage List. Moreover, about the costal area in which our sample test is, the final targets are related to the conservation, protection, management and planning of exceptional, ordinary, and degraded landscapes with particular reference to typical natural landscapes such as rivers, lakes, hills, mountains, coastal and rural landscapes, forestry and agro-pastoral, not to mention historic, rural, urban, industrial and infrastructure sectors (Table 3).
The objectives identified are also related to the government of the processes of urbanization and abandonment of the territory and to the preservation of material cultural values and intangible values such as the traditions and history of the Region.
The general objective was subdivided into specific objectives, as shown in the table, in which these objectives were finally associated with landscape quality directions that indicate policies to adopt and those who have an interest in achieving these objectives, as well as the measures required to adapt the urban planning instruments to the indications of the new Regional Landscape Plan.
Therefore, the landscape quality targets in this area aim to safeguard the surviving heritage in the area, to recover and improve the landscapes altered and degraded by human activity and to define quality standards.
Table 3. The landscape quality aims declined for the five resources systems in the sample area
Resources Systems |
General aims |
Specific aims |
||
|
Physical-Environmental System |
1 |
Promote the preservation of the integrity of areas of high naturalness and high ecosystem value |
1.1 |
Safeguard geological-geomorphological systems with high integrity (geological formations, ravines, cliffs, crags) |
|
1.2 |
Safeguard protected areas and areas of high environmental value such as those covered by the Nature 2000 Network |
|||
|
1.3 |
Safeguard and improve environmental functionality of river and lake systems of Molise |
|||
|
1.4 |
Safeguard and rebuild coastal marine habitats of Molise |
|||
|
1.5 |
Safeguard woods and forests of mountainous and hilly areas of Molise |
|||
|
1.6 |
Redevelop and redesign the coastal landscapes of Molise |
|||
|
Landscape-Visual System |
2 |
Promote improved integration of landscape and the quality of infrastructures |
2.1 |
Define territorial and landscape quality standards in the settlement of new network infrastructure |
|
2.2 |
Define territorial and landscape quality standards in the settlement of new energy infrastructure |
|||
|
2.3 |
Define territorial and landscape quality standards in the settlement of new productive activities |
|||
|
Historical-Cultural System |
3 |
Promote the preservation of cultural values |
3.1 |
Preserve cultural value and witnesses of settlements and historical manufacts |
|
3.2 |
Preserve cultural value of traditional rural buildings |
|||
|
3.3 |
Preserve the visible cattle-tacks residual |
|||
|
3.4 |
Redevelop the historic rural landscapes |
|||
|
Agricultural-Productive System |
4 |
Promote the conservation of agricultural landscapes |
4.1 |
Develop the agricultural landscape of Molise, recognize and promote its social functions |
|
4.2 |
Preserve open landscapes of the reclamation as a characteristic aspect of identity of coastal landscape of Molise |
|||
|
4.3 |
Redevelop the agricultural landscape of Molise |
|||
|
Demographic-Touristic System |
5 |
Promote the improvement of the quality of the settlements |
5.1 |
Improve quality of urban settlements and their environmental performance, for greater well-being of the population |
|
5.2 |
Redevelop degraded contemporary urbanization landscapes |
|||
|
5.3 |
Improve urban quality and touristic settlements |
|||
|
5.4 |
Improve urban quality of agricultural and productive settlements |
|||
|
5.5 |
Improve soft mobility quality (walking, cycling, trekking on horse) and its interconnection with traditional mobility |
|||
COMMENT 5
- It is important to present the recommendations for future studies in the end of the manuscript.
Reply 5
Thank you for your suggestions. Recommendations have been added in the conclusions.
Coastal planning is a central aspect of this study. In fact, the most critical elements of the demographic and economic development of the regional realities concerned converge on the coast areas. The government of the territory can be implemented through an intervention perspective that concerns a plurality of characters inherent in natural and artificial resources to guarantee protection of environmental and territorial peculiarities. The analysis and comparison between objectives, plan strategies, environmental compatibility criteria, environmental reference indicators and monitoring methods are all parameters that come into play but they are not sufficient. Therefore this case study is intended to be a contribution to the deepening of the relationship between territorial planning and landscape planning, since the urbanization processes of coastal areas inevitably influence even the innermost territories; also affected by related economic pressures and tourism development that endangers the integrity of the most fragile environmental and landscape contexts. The analysis carried out demonstrates the need to set up a methodology for the landscape which, although starting from data collection based on a homogeneous approach, in reality must always be adapted to local conditions, even within the same region. The final goal is to create vital landscapes.
This study was performed in two different scales: at the territorial level and at the local level, that is to say municipal level. At the territorial level, the indications are intended to address and propose planning actions. At local level, the municipal plans should act in accordance with the landscape plan suggestion: within the areas mentioned above, different situations are identified (in relation to the features and values of the natural environment and human actions) that require specific norms and guidelines. The overlap of the current and forecast condition also allowed us to have an immediate (and quantifiable) perception of dangerous, intense, extensive and persistent modifications, especially where there is neglect and poor preservation.
The future improvement of the research will be the suggestions for interventions that municipal administrations could promote, as is already happening in some other regions, as an active tool for the protection of the landscape within their municipal planning tools, in compliance with what is defined by the Regional Landscape Plan.
Reviewer 3 Report
I consider the subject addressed in the paper "Landscape Features of Costal Waterfronts: Historical Aspects and Planning Issues" it is very important in the context of the influences of urban developments and the effects they have on the landscape.
My observations are as follows:
- The structure of the paper makes it difficult the correct and easy perception of the research done by the author;
- the summary should be much more conclusive, highlighting the purpose, the methods used and the results of the paper;
- at the end of the Introduction the author describes the structure of the work and states that "(Section 1) has described the local and the wider context" - this section should be included in the Materials and Methods;
also in Material and Methods the author introduced "a review of methodological approaches for landscape analyses" but the results of this review are detailed in the introduction; The conclusions are missing!I recommend complete restructuring of the paper!
Author Response
The Author would like to thank the Referees for their comments. The manuscript has been modified following the suggestions, when deemed appropriate (new parts are in green). In this document, please find the details of my responses.
The Author.
REFEREE 2
Comment 1
the summary should be much more conclusive, highlighting the purpose, the methods used and the results of the paper
Reply 1
The abstract has been rewritten to be more conclusive, as suggested.
Comment 2
at the end of the Introduction the author describes the structure of the work and states that "(Section 1) has described the local and the wider context" - this section should be included in the Materials and Methods;
Reply 2
The text has been restructured
Comment 3
also in Material and Methods the author introduced "a review of methodological approaches for landscape analyses" but the results of this review are detailed in the introduction; The conclusions are missing!
Reply 3
Ok. Corrections made and conclusions created.

Reviewer 4 Report
The author provided a very detailed analysis on the landscape features of the waterfront coasts, and seemingly worked a lot on this topic. However, I have several questions, notes, and suggestions before it can be accepted for publication.
Abstract
I suggest to restructure a little bit this section:
I would leave the first 5 rows start with the “In the panorama …” sentence but rephrase it to have a topic sentence, placing the problem, and highlighting why is it important and then the author should focus what she had conducted as a research and what are the findings; partly this content can be found here, but the phrasing is rather general – more specific facts are needed finally, it would be good to have a closing sentence with referring the possible users of the results and readersIntroduction
I have the same problem with the introduction, as with the abstract: classically, we should see a background of the topic, and it is a good point to apply a inductive approach: from global to local, i.e. how large is this is topic, what areas are concerned, and why. Only seaside cities or rivers and lakes are also involved (only the level of the problem and not in this research)?L26-27: it seems a bit of “in medias res”, and I also do not understand why would potential urban quality of waterfronts guarantee anything; although here we found some background of this research, it is also general: in what terms and level? I see that the manuscript is about the Adriatic part of Italy, but it was in the Abstract, and it should be explained in details here … or to be more specific, not here, because the beginning should be what I suggested in #1
L32-36 is still too early, it needed to be moved and rephrased to the end of Introduction
L37-43: it more less good to be good for second paragraph … but still in a general form, Molise Region is rather specific problem, and narrows the potential readers; it is only a question of phrasing: there are general problems with the waterfronts, and there specific ones which are only true in Italy and this region – I suggest to separate this: at the beginning at least 2 paragraphs needed which explain the problem in general, globally.
I do not suggest the subsections within the introductions, it is better to have shorter but more focused introduction. Now it is wordy, a but of redundant, and I do not see the necessity of this long version. One third of this text would be enough, maybe a bit more. Please consider it due to the following reasons:
in this form it is very specific, contain names of places being not familiar to the non-local readers (e.g. Termoli Industrial Centre, Campomarino, Montenero di Bisaccia, Apulia, Lucania) – please consider that these places can be familiar to Italians, maybe to Europeans but this journal is an international one, and there will be readers from Africa, Asia, America, Australia and those never meet this cities, districts please find why this city/region is appropriate for a detailed analysis, how the results can be extrapolated to other cities/regions – if you cannot, the results stay at local level and will bo not in the interest to be cited which is against the journal’s policy I found what I intended to see at beginning as a starting thought, more or less in the subsection (it also means that why is there a long text before that?) However, it is also long and wordy … I am sorry to say but in its present form looks like the author wanted to share all of her thoughts and have too many subjects The whole introduction should be reduced to 4-6 paragraphs (100-120 rows) with the aims. Last paragraph is for the aims and the last but one for the research gap: what is the novelty of this study from the scientific research? Aims are not equal with the list of the contents, I suggest to rephrase it. Maybe a hypothesis-like approach would be the best, as it can also be used in the discussion to be confirmed.Methods
Apparently, the author (or considering the amount of work, the team behind her) conducted a nice work, I see that she was working on a development plan together with the local authority to have a better environment, to provide a background (database in GIS environment) for the future tasks and planning. However, we do not have the information on the conducted work … I suggest some points to improve this part:
a subsection (e.g. 2.1) is needed about the study area: there are lots of information in the introduction which can be used here, with those information which important to know when someone wants to understand the intent of the author the map (Fig. 1) is also important to move here what is New Landscape Plan and what is the author’s contribution to it and how? Now we only the background, but not what really happened, and how (how the data was arranged to a database?) … why is it a scientific problem? what is the novelty in it? what is the software environment for the GIS database? how satellite images can support the waterfront-related decisions? how the three types of grids were determined? what is the methodology?Methodology is more important than to read the lot of information in the introduction: the procedure should be detailed and should be enough to replicate … now it is not a detailed description, rather general. A draft figure of workflow, and the descriptions are important. Also, Table 1 is an important one, and a very detailed, but I cannot evaluate it in the context of the text: how resource systems contribute to the methodology? what is the outcome? just a database or it gives an evaluation, too?
Results
It is confusing because I had some answers for my previous questions here, in the results.
L406-478: it is methodology
It is very hard to judge the quality of the work without seeing the whole procedure, although moving the signed text to the methodology (please make subsections) improve the understanding, I do not see how these grids and the database have the contribution in identifying sensitivity. I am sorry to say but this is crucial, and without quantified and statistically analysed results, I cannot accept the manuscript.
Discussion
This is not discussion because it lacks the most important elements of the discussions:
results should be explained here … but results are not quantified, however, the author promised quantification at the beginning results should be placed in the previous research findings: it similar or there are contradictions? and if the latter, why?I, as an author, do not like when I get negative critics, and I do not like when reviewers do not tell what are their main issues. Negative critics do not make an author happy, neither me. However, I tried to help, and I hope she does not feel that I did not like her work, and directly wanted to refuse all the manuscript. The case is, as I see, she had a large amount of work in the manuscript, but the writing is not succeeded so well. I tried to give the main points what are the most important steps to be takes, what should be rephrased – no other high quality journal will accept it without a complete rethinking of it. I summarize it again:
a new introduction is needed: 4-6 paragraphs, and please focus on the subject and keep in mind that non-Italian readers also should be targeted, the final aim is not publish a paper but to gather citations; if the scope is too narrow, people will not read it methods should be restructured, and avoid the long wordy descriptions, instead short and relevant information is needed, and the aim is to provide a description which ensure the full-understanding what happened, and if they interested, the replication; now it is too general, and there is no description what can be quantified and how, results should be quantified, and there should e some statistical quantification (Past Tense, and there are no citations) discussion should explain the results, and how the results act in the mirror of previous studies – there should be some previous research (general statements are in Present Tense) conclusions: now it is completely missing – it is a short section enhancing the novelties, i.e. the new findings; I did not see it in the results to be highlighted …Author Response
The Author would like to thank the Referees for their comments. The manuscript has been modified following the suggestions, when deemed appropriate (new parts are in green). In this document, please find the details of my responses.
The Author.
REFEREE 3
Comment 1
Abstract
I suggest to restructure a little bit this section:
I would leave the first 5 rows start with the “In the panorama …” sentence but rephrase it to have a topic sentence, placing the problem, and highlighting why is it important and then the author should focus what she had conducted as a research and what are the findings; partly this content can be found here, but the phrasing is rather general – more specific facts are needed finally, it would be good to have a closing sentence with referring the possible users of the results and readers
Reply 1
The abstract has been rewritten, with a closing sentence with referring the possible users of the results and readers.
Comment 2
Introduction
I have the same problem with the introduction, as with the abstract: classically, we should see a background of the topic, and it is a good point to apply a inductive approach: from global to local, i.e. how large is this is topic, what areas are concerned, and why. Only seaside cities or rivers and lakes are also involved (only the level of the problem and not in this research)?
L26-27: it seems a bit of “in medias res”, and I also do not understand why would potential urban quality of waterfronts guarantee anything; although here we found some background of this research, it is also general: in what terms and level? I see that the manuscript is about the Adriatic part of Italy, but it was in the Abstract, and it should be explained in details here … or to be more specific, not here, because the beginning should be what I suggested in #1
L32-36 is still too early, it needed to be moved and rephrased to the end of Introduction
L37-43: it more less good to be good for second paragraph … but still in a general form, Molise Region is rather specific problem, and narrows the potential readers; it is only a question of phrasing: there are general problems with the waterfronts, and there specific ones which are only true in Italy and this region – I suggest to separate this: at the beginning at least 2 paragraphs needed which explain the problem in general, globally.
I do not suggest the subsections within the introductions, it is better to have shorter but more focused introduction. Now it is wordy, a but of redundant, and I do not see the necessity of this long version. One third of this text would be enough, maybe a bit more. Please consider it due to the following reasons:
in this form it is very specific, contain names of places being not familiar to the non-local readers (e.g. Termoli Industrial Centre, Campomarino, Montenero di Bisaccia, Apulia, Lucania) – please consider that these places can be familiar to Italians, maybe to Europeans but this journal is an international one, and there will be readers from Africa, Asia, America, Australia and those never meet this cities, districts please find why this city/region is appropriate for a detailed analysis, how the results can be extrapolated to other cities/regions – if you cannot, the results stay at local level and will bo not in the interest to be cited which is against the journal’s policy I found what I intended to see at beginning as a starting thought, more or less in the subsection (it also means that why is there a long text before that?) However, it is also long and wordy … I am sorry to say but in its present form looks like the author wanted to share all of her thoughts and have too many subjects The whole introduction should be reduced to 4-6 paragraphs (100-120 rows) with the aims. Last paragraph is for the aims and the last but one for the research gap: what is the novelty of this study from the scientific research? Aims are not equal with the list of the contents, I suggest to rephrase it. Maybe a hypothesis-like approach would be the best, as it can also be used in the discussion to be confirmed.
Reply 2
Thank you for your suggestion. The introduction has been rewritten.
Comment 3
Methods
Apparently, the author (or considering the amount of work, the team behind her) conducted a nice work, I see that she was working on a development plan together with the local authority to have a better environment, to provide a background (database in GIS environment) for the future tasks and planning. However, we do not have the information on the conducted work … I suggest some points to improve this part:
a subsection (e.g. 2.1) is needed about the study area: there are lots of information in the introduction which can be used here, with those information which important to know when someone wants to understand the intent of the author the map (Fig. 1) is also important to move here what is New Landscape Plan and what is the author’s contribution to it and how? Now we only the background, but not what really happened, and how (how the data was arranged to a database?) … why is it a scientific problem? what is the novelty in it? what is the software environment for the GIS database? how satellite images can support the waterfront-related decisions? how the three types of grids were determined? what is the methodology?
Methodology is more important than to read the lot of information in the introduction: the procedure should be detailed and should be enough to replicate … now it is not a detailed description, rather general. A draft figure of workflow, and the descriptions are important. Also, Table 1 is an important one, and a very detailed, but I cannot evaluate it in the context of the text: how resource systems contribute to the methodology? what is the outcome? just a database or it gives an evaluation, too?
Reply 3
As suggested, this paragraph has been restructured, with the inclusion of the workflow.
Comment 4
Results
It is confusing because I had some answers for my previous questions here, in the results.
L406-478: it is methodology
It is very hard to judge the quality of the work without seeing the whole procedure, although moving the signed text to the methodology (please make subsections) improve the understanding, I do not see how these grids and the database have the contribution in identifying sensitivity. I am sorry to say but this is crucial, and without quantified and statistically analysed results, I cannot accept the manuscript.
Reply 4
As suggested, this paragraph has been restructured, with subsections in the methodology.
Comment 5
Discussion
This is not discussion because it lacks the most important elements of the discussions:
results should be explained here … but results are not quantified, however, the author promised quantification at the beginning results should be placed in the previous research findings: it similar or there are contradictions? and if the latter, why?
I, as an author, do not like when I get negative critics, and I do not like when reviewers do not tell what are their main issues. Negative critics do not make an author happy, neither me. However, I tried to help, and I hope she does not feel that I did not like her work, and directly wanted to refuse all the manuscript. The case is, as I see, she had a large amount of work in the manuscript, but the writing is not succeeded so well. I tried to give the main points what are the most important steps to be takes, what should be rephrased – no other high quality journal will accept it without a complete rethinking of it. I summarize it again: a new introduction is needed: 4-6 paragraphs, and please focus on the subject and keep in mind that non-Italian readers also should be targeted, the final aim is not publish a paper but to gather citations; if the scope is too narrow, people will not read it methods should be restructured, and avoid the long wordy descriptions, instead short and relevant information is needed, and the aim is to provide a description which ensure the full-understanding what happened, and if they interested, the replication; now it is too general, and there is no description what can be quantified and how, results should be quantified, and there should e some statistical quantification (Past Tense, and there are no citations) discussion should explain the results, and how the results act in the mirror of previous studies – there should be some previous research (general statements are in Present Tense) conclusions: now it is completely missing – it is a short section enhancing the novelties, i.e. the new findings; I did not see it in the results to be highlighted …
Reply 5
As suggested, this paragraph has been restructured (discussion and conclusions), referring previous researches, novelties and future possibilities.

Reviewer 5 Report
This study investigates the relationship between production and settlement structures along the coasts, through a case study carried out on the Italian coast of the Adriatic Sea, affected by a strong increase in buildings right along the shoreline.
The manuscript is well organized; the reader easily reads the text and figures. Hence, the paper can be judged to add knowledge in its topic and it adheres to the “Sustainability” standards.
The author just should check the origin of the constraint of 300 m from the coast, which probably dates back to a law of 1985 and not of 1939.
Furthermore, it seems preferable to distinguish between a "Discussion" and a "Conclusions" section.
Author Response
The Author would like to thank the Referees for their comments. The manuscript has been modified following the suggestions, when deemed appropriate (new parts are in green). In this document, please find the details of my responses.
The Author.
REFEREE 4
Comment 1
The author just should check the origin of the constraint of 300 m from the coast, which probably dates back to a law of 1985 and not of 1939.
Reply 1
It should be corrected. The sentence was longer and cited both laws. When I narrowed the phrase I was wrong about the date, but certainly the constraint concerning 300 mt from the coast was introduced by the law 431/1985. To be honest the Molise Region has a legal exception since the 300 meters are completely flooded and then the constraint for the Molise coast has been redefined by the Region with resolution GR 272 7 February 1996 that for the distance from the shoreline states: “for A2N1 areas the compatible uses are permitted within 40 meters from public infrastructures (roads, railways and parking lots) running parallel to the shore line”.
Comment 2
Furthermore, it seems preferable to distinguish between a "Discussion" and a "Conclusions" section.
Reply 2
Thank you for your suggestion. Conclusions paragraph has been created.

Reviewer 6 Report
The manuscript titled "Landscape Features of Costal Waterfronts: Historical Aspects and Planning Issues", according to what is stated in the abstract, aims at exploring the relation between factors affecting urbanized coastal areas by means of a method implemented in the plan-making process concerning the regional landscape plan for Molise, an Italian region.
This article does have potential, in that the topic is worth investigating, and the method developed is interesting. The discussion and conclusions highglight relevance and usefulness of the study. However, the manuscript shows various weaknesses for which major reworking is needed. Next, I will try to highlight some weak areas and provide suggestions for improvement.
1) In my opinion, the weakest point of the manuscript is its organization. It lacks a neat structure, which makes it hard for the reader to follow the author's line of reasoning. I would recommend having a close look at the journal's "instruction for authors" webpage (https://www.mdpi.com/journal/sustainability/instructions) which does provide good advice on how to structure properly a research article. In the following paragraphs, for the first three sections of the manuscript, I will first write what each section should do (italicized part) and next provide some recommendations.
1a) The introduction should set the scientific context for the research and provide the reader with the debates to which the study adds something, identyfing the gap in the liteature that the study wants to address, form which the the research question should logically follow; moreover, it should also explain why this research is relevant and novel.
In the paper, the introduction starts with the aim (lines 26-31) and provides a research goal that is quite different to the one stated in the abstract. Next, two paragraphs justify why the literature reviewed is Italian and what follows is about laws concening landscape assets and goods, then about the landscape planning framework and examples from other Italian regions, and finally about previous projects carried out by the author's research group. Hence, the introduction does not succeed in setting the scientific context on which the study is grounded; for this reason, the author is advised to revise this section as follows:
- do not start the intro by stating what the aim is - this should stem from the literature.
- first present the literature and next, if needed, provide reasons for only looking at national debates.
- after providing a flavour of the literature debates on coastal areas (developed/urbanized), do state what the research gap is - hence, what the literature has until now failed to address.
- next, explain what the study does to fill in this gap, i.e., what the research question or the aim of the paper is, and what its novelty is
- legal and planning framework should be kept short, as this is not a technical report
- do use here some of the material which is currently misplaced in section 2.1.
1b) Materials and Methods: this section could be split into two subsections: one about the case study (currently lines 316-357) and one about the methodology (currently Figure 1 + sections 2.2, 2.3 and 2.4). The subsections could be even three, if the author deems that data gathered deserve a separate slot.
The largest part of this section (section 2.1 until line 316) is actually a literature review, which should better be moved to the introduction.
Concerning the methodology (Table 1), the rationale behind setting up four main systems and choosing which elements are to be included in each system has not been explained, which makes it hard for the reader to fully understand the author's choices. For instance, out of the many questions this table raises: why is slope a detrattore (incidentally, "detractor" is not the correct English word the author wants: https://www.oxfordlearnersdictionaries.com/definition/english/detractor?q=detractor ) if we consider the way the author defined "detractors"in lines 361-362 ("elements that invaded [sic] the territory creating uncontrolled development")? why are treatment plants listed both in the second row (landscape system) and in the fourth one (demography and tourism)? why all of the age classes are regarded as values (fourth row)? what is a "reclamation consortium"?
3) Results: this section should present the outcomes of the analysis without analysis or inferences.
Instead, lines 414-485 + lines 497-503 ("Their selection ... indicators") are about the methodology, since the author here explains what has been done, on which data, etc, while lines 515-526 are about the case study and lines 527-558 provide additional info, which does not appear to be directly linked to the analysis carried out but which, nonetheless, could strengthen the discussion. Hence, only lines 486-496 really concern the results.
The author is advised to move the above mentioned parts to their appropriate sections and to add some flesh to the results.
4) The English needs a thorough revision. A number of sentences appear to be literally translated from the author's native language (e.g.: what are the "eight floors" in line 402? what does the author mean by "hydraulic setup" in Table 1?), hence the help of a native speaker or proofreader would greatly help the author better convey the intended meaning.
5) As for satellite images, please note that sensors are not photographic cameras, hence the phrase "having been photographed using the MIVIS" is incorrect.
Author Response
The Author likes to thank the Referees for their comments. The manuscript has been modified following the suggestions, when deemed appropriate (some paragraphs are reorganized: only new parts are in grey). In this document, please find the details of my responses.
The Author.
COMMENT 1
1a) The introduction should set the scientific context for the research and provide the reader with the debates to which the study adds something, identyfing the gap in the liteature that the study wants to address, form which the the research question should logically follow; moreover, it should also explain why this research is relevant and novel.
In the paper, the introduction starts with the aim (lines 26-31) and provides a research goal that is quite different to the one stated in the abstract. Next, two paragraphs justify why the literature reviewed is Italian and what follows is about laws concening landscape assets and goods, then about the landscape planning framework and examples from other Italian regions, and finally about previous projects carried out by the author's research group. Hence, the introduction does not succeed in setting the scientific context on which the study is grounded; for this reason, the author is advised to revise this section as follows:
- do not start the intro by stating what the aim is - this should stem from the literature.
- first present the literature and next, if needed, provide reasons for only looking at national debates.
- after providing a flavour of the literature debates on coastal areas (developed/urbanized), do state what the research gap is - hence, what the literature has until now failed to address.
- next, explain what the study does to fill in this gap, i.e., what the research question or the aim of the paper is, and what its novelty is
- legal and planning framework should be kept short, as this is not a technical report
- do use here some of the material which is currently misplaced in section 2.1.
Reply 1
Thank you for your suggestions. The paragraph was re-organized..
COMMENT 2
1b) Materials and Methods: this section could be split into two subsections: one about the case study (currently lines 316-357) and one about the methodology (currently Figure 1 + sections 2.2, 2.3 and 2.4). The subsections could be even three, if the author deems that data gathered deserve a separate slot.
The largest part of this section (section 2.1 until line 316) is actually a literature review, which should better be moved to the introduction.
Concerning the methodology (Table 1), the rationale behind setting up four main systems and choosing which elements are to be included in each system has not been explained, which makes it hard for the reader to fully understand the author's choices. For instance, out of the many questions this table raises:
why is slope a detrattore (incidentally, "detractor" is not the correct English word the author wants: https://www.oxfordlearnersdictionaries.com/definition/english/detractor?q=detractor ) if we consider the way the author defined "detractors"in lines 361-362 ("elements that invaded [sic] the territory creating uncontrolled development")? why are treatment plants listed both in the second row (landscape system) and in the fourth one (demography and tourism)? why all of the age classes are regarded as values (fourth row)? what is a "reclamation consortium"?
Reply 2
In my first version, the literature review was located in the introduction but other referees suggested changing the layout. However, now the paragraph has been further revised.
Regarding, then, the use of the term "detractor" I would like to say that, after long dissertation (which is accounted for in the publications cited in the bibliography), in common agreement with the other researchers of the European Interreg project conducted in past years (and also mentioned in the bibliography), we coined the term, which in reality is not completely verifiable also in Italian, officially introducing it into the project and giving the explanation that we have also reported in this text. In fact, in the first version it was put in quotes and now I put it back in quotes (both “value” and “detractor”).
COMMENT 3
3) Results: this section should present the outcomes of the analysis without analysis or inferences.
Instead, lines 414-485 + lines 497-503 ("Their selection ... indicators") are about the methodology, since the author here explains what has been done, on which data, etc, while lines 515-526 are about the case study and lines 527-558 provide additional info, which does not appear to be directly linked to the analysis carried out but which, nonetheless, could strengthen the discussion. Hence, only lines 486-496 really concern the results.
The author is advised to move the above mentioned parts to their appropriate sections and to add some flesh to the results.
Reply 3
The paragraph was revised.
COMMENT 4
4) The English needs a thorough revision. A number of sentences appear to be literally translated from the author's native language (e.g.: what are the "eight floors" in line 402? what does the author mean by "hydraulic setup" in Table 1?), hence the help of a native speaker or proofreader would greatly help the author better convey the intended meaning.
Reply 4
A further English language check (as in previous versions), has been carried out by a native speaker, who collaborates with us and who is constantly updated on the progress of our research. If some sentences are not clear, please give us specific examples.
For hydraulic setup: it refers to the hydro-geological structure
COMMENT 5
5) As for satellite images, please note that sensors are not photographic cameras, hence the phrase "having been photographed using the MIVIS" is incorrect.
Reply 5
New sentence:
It was updated for this work, having proceeded to purchase a series of photographic and satellite images (Landsat TM, Spot and Quick Bird) from mid 90’s to present day; particular attention was paid to MIVIS data, appropriately corrected radio-metric and above all geo-coded (orthorectified)) and georeferenced.
Round 2
Reviewer 1 Report
The Author did not successfully address all previous comments and concerns.
The manuscript still requires an extensive revision of English in order to be publishable.
The Author indicated that in a new manuscript all changes are highlighted in green, while there are parts not highlighted that are new as compared to the previous version. This is a shortcoming for the transparency of this peer-review process.
Most important, specific issues which were ignored by the Author:
Re-writing the abstract to emphasize the scope and goal of the study, (In fact, very little was modified from its previous form).
Comments 11, 12 and 13: if the official documents are not successfully describing all necessary elements of the landscape with its values, why select them? This looks like a flawed methodology.
Author Response
The Author likes to thank the Referees for their comments. The manuscript has been modified following the suggestions, when deemed appropriate (some paragraphs are reorganized: only new parts are in grey). In this document, please find the details of my responses.
The Author.
REFEREE 1
COMMENT 1
Re-writing the abstract to emphasize the scope and goal of the study, (In fact, very little was modified from its previous form).
Reply 1
The abstract was re-organized.
New abstract:
Abstract: This paper investigates the relationship between different factors that impose on the productive and settlement structures on coastal areas, through an analysis carried out on the Italian Adriatic Sea coast. In the panorama of medium and small size cities, the relationship between the city, the territory and the sea very often plays an important role. The main issue of this article is to expose a methodology developed for the definition of landscape quality objectives in the planning of the coast of a region in Southern Italy, Molise. Effort was concentrated on the creation of a territorial survey matrix that could be exploited by local authorities. In drawing up the criteria on which to base the New Regional Landscape Plan, this study provided for the recognition of the identifying matrices for landscape interpretation, creating a database organized in five resource systems. For each resource system three basic grids were created: each of them collects and processes different information series. These three grids were useful for defining the new protection that is proposed for the sample area. Different conditions emerge in this area, in which two coastal strips have been identified, to the east and to the west of the historical centre.
COMMENT 2
Comments 11, 12 and 13: if the official documents are not successfully describing all necessary elements of the landscape with its values, why select them? This looks like a flawed methodology.
Reply 2
The methodology, created for the New Landscape Plan of the Molise region, has already been trialled in other areas of the region; in this work it has been applied to the coastal area around the capital of the province, Termoli. It means that the general asset of the methodology must necessarily include all possible values. In some cases these values are zero.
As regards comment 11, it has already been explained: the dunes in the case of the entire Molise region cannot be considered an element of environmental value since they have disappeared. They are only considered as "residual areas" since they are considered important in the landscape-visual system.
Also in the case of comment 12, it was explained that extensive agriculture in the region in question should not be considered a negative element since it does not reach levels of harmfulness to the environment.
Also for comment 13 there was replied. It should perhaps be added that the infrastructures in the region are never a value since as they pass through it and constitute a strong negative element.
In any case, as requested by another referee, the five resource systems are explained in detail, even if they are not the fulcrum of this work.
Reviewer 3 Report
The structure of the work is substantially improved, much easier to follow. The sections are correlated with each other. A well documented work.
Author Response
Comment
The structure of the work is substantially improved, much easier to follow. The sections are correlated with each other. A well documented work.
Reply
Thank you for your comments and your final evaluation.
Reviewer 4 Report
I appreciate that the author performed the revision within a short time, and accepted most of my suggestions. However, now it still cannot be accepted, as this new version which should had been the submitted one for the first time, from this point I can begin the real review work.
General comments
Now, this form is relevantly better than the previous one: it has a better structure and reasonable length and most of the unnecessary parts are removed. However, some very important points of my previous review are not handled, and before acceptance, it should be thoroughly revised. Quantified results are completely missing, and methodology is still not understandable. Accordingly, discussion and conclusions are also flawed.
Specific comments
Abstract
Now it better but not good. I know that the length is maximized in words, but the beginning is long (maximum 1-2 sentences), there is nothing about the methodology and results (I mean quantified results, coming from the analysis).
Introduction
L26-33: classically, a paper starts with a general approach, and not with the performed analysis and the aims. This part supposed to be the last paragraph in the introduction.
As in my first review, I still would like to see this general beginning, giving a worldwide picture why this topic is important.
L39-42: it is not a good part, we do not know what “literature analyzed” refers to: to previous works, or this one? I guess, the landscape planning varies by countries, so Italy should be different from the others. However, without pointing on the facts, it only general, which does not seem as a scientific contribution or statement.
The next three paragraphs (government and territory, landscape, landscape heritage) are existing terms in other countries, in scientific literature. These points should be explained what is the difference in the Italian terminology. This way, if I “translate” this part the text says: Italy has its own nomenclature, regulations, and planning practice and this is the uniqueness of the submission. However, the author should keep in mind that Sustainability is an international journal, and has a worldwide publicity. In this form, this whole part L39-L68 concerning only the Italian specialities, cannot be accepted.
L68-88: this paragraph is a bit descriptive and too specific (only Italian issues are presented), the style is not of scientific: e.g. “interesting document”, and the content is rather a presight of some upcoming topics, with methods, introduction and in the same time, lacks the facts.
… and in the next paragraphs the topic is explained why is it important, but without a straight logic. Even the aims are just the description of the next sections, however, I suggested that this is not a good approach: if possible, please use hypothesis, if not, try to create scientific questions. Just take a look at some previous papers who the authors phrased the aims.
L89: I have never seen a question in this form in scientific paper
L90: “undoubtedly includes” – I would remove this type of phrasing
L105: please avoid the one-sentence paragraphs, and here is a small advice how a paragraph should look like:
the first sentence is a “topic sentence”, it specifies the content of this little part then, in 8-10 lines the topic is detailed the last sentence is “closing sentence” which gives a small summary or conclusionSo, to sum it up, please follow this structure:
1st paragraph: general introduction of the topic, why is it worth to read? why is a global issue in all over the world? what consequences can be identified? what happens if this issue is not handled or even planner do not recognize it as a problem? 2nd paragraph: what are the methods of identifying this problem? how easy to implement the procedures regarding the methods and the data requirements? how reliable are these methods?These 2 paragraphs require a comprehensive literature review, because now it is completely missing. The author has to find the all the relevant previous studies and should involve them into her line of thoughts.
3rd paragraph: the author can focus on Italy and shortly refer to the unique issues of Italian questions, but only in comparison with other countries, and the existing literature 4th paragraph: what happened until now and what is missing from the existing studies (even in the world, not just Italy), this should focus on the novelty of this submission 5th paragraph: aims, as I suggested previouslyMethods
L122-130: Do not forget that the journal is interested in scientific approaches, because the practice usually uses old well-working techniques, but works well in a given task, however, scientifically sometimes can questioned …
L123: “a new stage” do not forget that readers do not know the previous stages
L126: what is the matrices of Landscape Units? What is a landscape unit?
L131-132: this is the style which is definitely does not fit to a scientific paper, please rephrase it in other way, and refer to the figure only in brackets
Now, I will not point on the sentences in details, but the structure is not good. I have the same issues with it like in the introduction: I am missing a straight logic, and the paragraphs have no meaning, paragraphs are mixed with elements of introduction, and ideas (not literally) to the discussion
Just some disturbing examples:
L159: the author says that there is an extensive literature of topic of waterfront, however, cites only one item … and this is definitely not belongs to methodology
The upcoming paragraphs (L165-353) intended to show the details, but it is not necessary here, maybe in the discussion, but not this descriptive way, instead with a comparison of the results of this manuscript: what is similar and what differs and what is the reason (only in a short way). Most of these paragraphs can be published as a review as can be valuable examples … but the simple descriptive parts should be rephrased in that new paper, too. The most important is to understand that this paper is an original submission and not a review. As a review it is not good too, but as a research paper the methodology should focus on the methods itself and the relating literature. … or it can be a chapter of a book …
L258-266 Bilbao effect can be interesting, but maybe not in the methodology
And regarding the methodology, I still do not understand the following points and have some missing parts :
the 2.1. section should be for the “Study area” where the author should give those information which is important to know to understand the waterfront issues (topography, environment, demography, tourism); the figure of study area should be placed here 2. should give the information of the applied methods, e.g. I do not know what is the mapping element? a plot/parcel? a district? a house? and does it differ in case of topography and demography? later we can see a term, “grid” but we know that there were 3 grids … nothing more: was it the unit? in what geometric resolution? the new figure helps in the understanding but is far from the ideal and contain phrasing errors, please check it with a native speaker how were the layers summarized? adding up? weighting? … this is just a next step what is not defined in the methods what is/can be the validation of this method? in L367 there is a referring of validation, but is just a word in its form, the main question is what is validated and how? what was the role of purchasing satellite images? just a small note that Landsat is free (30 m fits to the aims?), Quickbird is not operational since 2015 how the satellite images were implemented? spectral indices, classification? … and the ever question: why, what was the purpose?L405-406: comparison with what and how (which method) and why?
Now the methods is long and should be completely rewritten.
Results
… I can say the same as previously: fragmented sentences and the structure can be more focused.
L414-485: this belongs to the methods section, but only in a shortened format, now it is wordy, and long touching too much topics which have relation with the topic itself but neither methodology and nor results
The rest of the text also contain methodology and results, but in the right form (methodology should be removed from here and results should organized in other form).
My suggestion would be the next:
as a rule, we can say that only those information counts which is quantified, i.e. in this manuscript nothing was quantified, only descriptive presentation can be found … this is not acceptable accordingly, the first step is to quantify the results, the author should know what … I would guess the factors which influence the judgement of the main topic, the waterfront topic and issues, such as topography, demographic/touristic pressure, etc ten, the compiled maps come: I think the grids, and then, with maps, diagrams, these results should be presented as a second subsection of the results then, I suggest some validation: find correlations or differences (hypothesis testing), i.e. statistical confirmation of relating factorsNow, the current content is not like an acceptable results’ section, it should be rewritten.
Discussion
As the results section is not appropriate, the discussion section also should be rewritten. However, at first, quantified results are needed. Then, quantified results should be explained and placed into the international literature, as a confirmatory analysis.
Conclusion were not in the right form, and the added text did not helped much. According to the quantified results, it should be rewritten.
Author Response
The Author likes to thank the Referees for their comments. The manuscript has been modified following the suggestions, when deemed appropriate (some paragraphs are reorganized: only new parts are in grey). In this document, please find the details of my responses.
The Author.
Total Reply
The abstract was re-organized.
About the other paragraphs, they were revised. Moreover I would like to underline (as included now in the text) that:
“In order to prepare the New Regional Landscape Plan (under Agreement between the Molise Region and the l.a.co.s.t.a. Laboratory (Director prof. D. Cialdea) of the University of Molise for the realization of the “New Regional Landscape Plan of Molise”, signed 11.02. 2011, REG. n. 11 del 28.02.2011., a table of the new landscape quality aims as listed in art. 135 of the National Code (L. 42/2004) was organized: it contains for each homogeneous area specific requirements and provisions, oriented to the conservation of the constituent elements, to the rehabilitation of compromised or degraded areas; to the protection of landscape features; to the identification of the lines for the development, with particular attention to the preservation of rural landscapes and sites included in the UNESCO World Heritage List. Moreover, about the costal area in which our sample test is, the final targets are related to the conservation, protection, management and planning of exceptional, ordinary, and degraded landscapes with particular reference to typical natural landscapes such as rivers, lakes, hills, mountains, coastal and rural landscapes, forestry and agro-pastoral, not to mention historic, rural, urban, industrial and infrastructure sectors. The objectives identified are also related to the government of the processes of urbanization and abandonment of the territory and to the preservation of material cultural values and intangible values such as the traditions and history of the Region.”
About the indicators, the methodology, created for the New Landscape Plan of the Molise region, has already been trialled in other areas of the region; in this work it has been applied to the coastal area around the capital of the province, Termoli. It means that the general asset of the methodology must necessarily include all possible values.
Reviewer 6 Report
I am happy with the improvements made in this revised version and the way the author handled the comments and suggestions provided in the first round.
Please note that tables 1, 2, and 3 are provided as low-image resolution, whose quality should be enhanced.
Some minor language glitches are still present; however, they do not prevent the reader from understanding the meaning of the text.
Author Response
Thank you for your suggestion. I enhanced the graphic resolution of the tables
Round 3
Reviewer 1 Report
The manuscript was to some extent corrected but presents a modest contribution to the field. Its level of novelty is quite modest: "The identification of resource systems (...) is considered the innovative element of the work.", there already are multiple frameworks of assessment of the landscapes at the regional scale, which are reliable and valid tools. It is not stated how the new one is better than the existing ones. Moreover, the classification of resources and detractors under each of the resource systems is quite controversial (as pointed out in previous comments). The controversy pointed out hasn't been successfully and convincingly explained in the manuscript as well as in the answers to the reviewer.
The paper in its current form is not publishable, as the methods and presentation of the findings have to be deeply re-considered by the Authors. Also, the level of scientific writing and organization of sections is not adjacent to current standards in academic writing.
Below some other comments that could assist the Author in re-submission of the manuscript in the future.
The title of a manuscript is misleading. It suggests the study involves different coastal waterfront, while in fact, it refers to one case of Molise region.
ABSTRACT:
The Author does not successfully describe the main research objectives and goals of this study. From the statement in the abstract it is unclear whether the study objective is to identify landscape planning priorities/ guidelines OR to evaluate the landscape quality with the new methodology. Five resource systems should be listed in the abstract. There should be a mention of the study findings in the abstract. What did this study accomplish? What are the contributions?
KEYWORDS:
List of keywords should be reconsidered, as "urban built environment" "identity values" "Natura 2000" does not sufficiently appear in the text.
INTRODUCTION:
International debate mentioned in line 24 should have a reference. Same applies to the "many international researchers" mentioned in line 28. In line 31, what does the expression "are evoked" refers to: is it to the current manuscript, or other international researchers? It should be rephrased. Other European cities mentioned in ln. 34-36 should be mentioned or a relevant reference should be added. Line 57 - what great events?. Line 122-126 - redundant.
METHODS:
Figure 1 - the five resource systems and 3 grids should be named in the figure. Line 139 - which figure?
RESULTS
Lines 284-291 - the site characteristic should be part of the Introduction. Lines 292 on - Still, as pointed in the first round of comments, these information does not belong to the results section. But more to Methodology. Figure 2. seems to belong to the Materials and Methods section too, not the Results, as they are already existing images, not ones developed in the course of the study.The result section is lacking the graphical representation of the results. If any.
DISCUSSION:
From line 469 on there is an extensive description of a New Regional Landscape Plan, which is not the result of this study. It is unclear why this description is placed in this section. More Discussion section is generally too extensive and chaotic, Author introduces a lot of new content and new figures, instead of focusing on the relevance of the findings of presented study. It does not refer to principles of academic writing, for example, summarised as:
"‘Discussion ‘ section should not exceed the sum of other sections (ıntroduction, material and methods, and results), and it should be completed within 6–7 paragraphs. Each paragraph should not contain more than 200 words, and hence words should be counted repeatedly. The ‘Discussion’ section can be generally divided into 3 separate paragraphs as. 1) An introductory paragraph, 2) Intermediate paragraphs, 3) Concluding paragraph."
Moreover, the Discussion section fails to link back to the Introduction, where all the case examples of European cities have been described.
CONCLUSIONS:
The conclusion lacks a clear identification of the implications for planning and design.
_______________________________
LANGUAGE:
The manuscript still doesn't read well when it comes to the English language correctness. It is peppered with grammatical and stylistic errors, or direct translations, which do not read well. Below some, (not all!) examples of mistakes detected:
[line 11 and 123] "The main issue of the paper.." - we should strive to publish papers without issues.
[ln. 50-52], the sentence especially its last part isn't clear.
[ln. 52] - Does the Author mean "In the area of the regeneration interventions..."?
Figure 1 - a typo "requalification" not "riqualification"
Author Response
Comment for the abstract
the study objective is to evaluate the landscape quality with the new methodology
“The main issue of this article is to expose a methodology developed for the definition of landscape quality objectives in the planning of the coast of a region in Southern Italy, Molise.”
Comment for the keywords
The Author thinks that should not need to change
Comment for the introduction
References related to the international debate are present in the text.
Comment for the methods
Five resource systems and 3 grids are named in the figure.
Comment for the results
The Author thinks that should not need to change
Comment for the discussion
The organization of this paragraph has been rewritten in accordance with the comments of other reviewers
About the English language, thank you for your suggestions. Some typing errors have been corrected
Reviewer 4 Report
I appreciate the efforts of the author and the work devoted to improve the manuscript, but it is still not the one which can be accepted.
- The introduction is wordy, stull too long and consist of irrelevant examples, lacks the logic I delineated in my previous review: all paragraphs starts with “starting sentence” and closed by some conclusion … between them, the author has the possibility to explain the topic. Now, it is still full of 1-2 sentence long paragraphs. Seems that the author cannot decide what is important and what is not.
- The author ignored some my pervious comments, such as L212-213 still consist of vague statements: Landsat cannot be purchased as it is free, Quickbird is not operational today, and without facts (exactly what images were used, from what dates, and the most important: why?) it is not a scientific description. In previous version there was no any mention of MIVIS data … suddenly it is here now … I can be convinced, but the author has to provide a scientific explanation for the questions I took (what, when, why).
- Actually, I would like to see maps of the inputs. Now, I (and the readers) cannot imagine how the input data compiled into a system. E.g. Within Demographic-touristic system what is the “Planning tools”? in its form it is meaningless. Furthermore, there is a detractor, presence of asbestos with five categories … why five? which is the worst (1 or 5)?
- Results is still about just “writing”. As I previously pointed on, it should be about the quantified results. Or to show how the system helped the planning. But now, it is still has no real improvement.
- L257-266 should be in a subsection of Study area (although I suggested to have, this subsection is still missing) – anyway, in a section like that, it can be clarified the Molise Region and Termoli nomenclature: what is the manuscript about exactly? Please make it clear
- L266-283: wordy paragraph with lots of irrelevant information
- L284-291: should be moved to Study area subsection
- L292-L364: it is also not results … and it is not the first time to mention it … it is a wordy description of the methods, BUT: methods is unreasonably long now, we know a lot about the methods, but important information is missing … readers will lost the line and also their interest
- remaining paragraphs belong partly to the methodology and also lacks any exact analysis … in this form I cannot accept it
- Please rephrase the whole methodology and result section … too long and we dot know what data is involved (Table 1 hardly gives enough information), results are not satisfy the standards of being results
- Fig 1 is more or less good, but the title is not common, the subscription is enough. An element which is new for me after the revisions is the “territorial survey” together with L141: please define what was the survey (does it mean field measurements, data collection from the observations, or geodetic survey)? How it was implemented into the methodology?
- Fig 2 is a big combination of everything: a part should be in the section of Study area subsection, b and c supposed to be the only result. Here appears the term, Master Plan, which was never defined previously, only mentioned several times. That is what I am talking about: too wordy text, but without exact nomenclature where terms are defined, and used in the whole submission.
- Discussion still lacks the explanations, it is written in the same style as the results, the descriptive text is continued, without placing the results into the previous findings of the international literature, and to provide explanations.
- Conclusions should be short and concise and should focus on the new findings of the submission. Now it is long and does not have a focus.
I have to admit that I surprised the style of the “Answers”. International journals require detailed answers from point by point, every note/comment/question should be addressed, and cannot be considered as fulfilled with a statement that everything is revised and can be found in the text. If the author would answer the comments, maybe would notice that most of my previous notes/suggestions etc. were ignored.
Author Response
Thank you for suggestions about the whole article. The Author tried to add the suggestions of all reviewers, who naturally have different opinions, and tried to find the best solution
In particular: Landsat and Quickbird images relating to previous years have been purchased by us in order to create diachronic maps.